# Iterated Reasoning with Mutual Information in Cooperative and Byzantine Decentralized Teaming

**Sachin Konan**[*]**, Esmaeil Seraj**[*]**, Matthew Gombolay**
Georgia Institute of Technology
Atlanta, GA 30332, USA
`{skonan, eseraj3}@gatech.edu, matthew.gombolay@cc.gatech.edu`

## Abstract

Information sharing is key in building team cognition and enables coordination and cooperation. High-performing human teams also benefit from acting strategically with hierarchical levels of iterated communication and rationalizability, meaning a human agent can reason about the actions of their teammates in their decision-making. Yet, the majority of prior work in Multi-Agent Reinforcement Learning (MARL) does not support iterated rationalizability and only encourage inter-agent communication, resulting in a suboptimal equilibrium cooperation strategy. In this work, we show that reformulating an agent's policy to be conditional on the policies of its neighboring teammates inherently maximizes Mutual Information (MI) lower-bound when optimizing under Policy Gradient (PG). Building on the idea of decision-making under bounded rationality and cognitive hierarchy theory, we show that our modified PG approach not only maximizes local agent rewards but also implicitly reasons about MI between agents without the need for any explicit ad-hoc regularization terms. Our approach, InfoPG, outperforms baselines in learning emergent collaborative behaviors and sets the state-of-the-art in decentralized cooperative MARL tasks. Our experiments validate the utility of InfoPG by achieving higher sample efficiency and significantly larger cumulative reward in several complex cooperative multi-agent domains.

## 1 Introduction

Information sharing is key in building team cognition, and enables agents to cooperate and successfully achieve shared goals (Salas et al., 1992). In addition to communication, individuals in high-performing human teams also benefit from the theory of mind (Frith & Frith, 2005) and making strategic decisions by recursively reasoning about the actions (strategies) of other human members (Goodie et al., 2012). Such hierarchical rationalization alongside with communication facilitate meaningful cooperation in human teams (Tokadli & Dorneich, 2019). Similarly, collaborative Multi-agent Reinforcement Learning (MARL) relies on meaningful cooperation among interacting agents in a common environment (Oliehoek & Amato, 2016). Most of the prior works on collaborative MARL are based on the maximum utility theory paradigm which assumes perfectly informed, rational agents (Guan et al., 2021). Nevertheless, even under careful handcrafted or machine learned coordination policies, it is unrealistic and perhaps too strong to assume agents are perfectly rational in their decision-making (Wen et al., 2020; Gilovich et al., 2002; Simon, 1997; Paleja et al., 2021).

To learn cooperation protocols, prior MARL studies are commonly deployed under Decentralized Partially Observable Markov Decision Processes (Dec-POMDP), in which agents interact to maximize a shared discounted reward. More recently, *fully-decentralized* (F-Dec) MARL was introduced (Zhang et al., 2018; Yang et al., 2020) to address the credit assignment problem caused by the shared-reward paradigm in conventional Dec-POMDPs (Yang et al., 2020; Sutton, 1985). In an F-Dec MARL setting, agents can have varying reward functions corresponding to different tasks (e.g., in a multi-task RL where an agent solves multiple related MDP problems) which are only known to the corresponding

---

[*]Co-first authors. These authors contributed equally to this work.

agent and the collective goal is to maximize the globally averaged return over all agents. Nevertheless, under an F-Dec setting, agents seek to maximize their own reward, which does not necessarily imply the maximization of the team long-term return since agents do not inherently understand coordination.

Recently, strong empirical evidence has shown that *Mutual Information* (MI) is a statistic that correlates with the degree of collaboration between pairs of agents (Trendafilov et al., 2015). Researchers have shown that information redundancy is minimized among agents by maximizing the joint entropy of agents' decisions, which in turn, improves the overall performance in MARL (Malakar et al., 2012). Therefore, recent work in MARL has sought to integrate entropy regularization terms as means of maximizing MI among interacting agents (Kim et al., 2020; Wang et al., 2019; Jaques et al., 2019). The formulaic calculation of MI relies upon the estimation of action-conditional distributions. In most prior work, agents are equipped with conventional state-conditional policies, and researchers employ techniques, such as variational inference, for estimating and optimizing an action-conditional policy distribution to quantify MI (Wen et al., 2019; Kim et al., 2020). However, agents are not explicitly given the ability to reason about their teammates' action-decisions and, instead, have to learn implicitly from sparse rewards or hand-engineered regularization and auxiliary loss terms.

**Contributions –** In this work, we propose a novel information-theoretic, fully-distributed cooperative MARL framework, called InfoPG, by reformulating an agent's policy to be directly conditional on the policies of its instantaneous neighbors during Policy Gradient (PG) optimization. We study cooperative MARL under the assumption of bounded rational agents and leverage action-conditional policies into PG objective function to accommodate our assumption. By leveraging the $k$-level reasoning (Ho & Su, 2013) paradigm from cognitive hierarchy theory, we propose a cooperative MARL framework in which naive, nonstrategic agents are improved to sophisticated agents that iteratively reason about the rationality of their teammates for decision-making. InfoPG implicitly increases MI among agents' $k$-level action-conditional policies to promote cooperativity. To learn collaborative behavior, we build InfoPG on a communicative fully-decentralized structure where agents learn to achieve consensus in their actions and maximize their shared utility by communicating with their physical neighbors over a potentially time-varying communication graph. We show the effectiveness of InfoPG across multiple, complex cooperative environments by empirically assessing its performance against several baselines. The primary contributions of our work are as follows:

1. We derive InfoPG, an information-theoretic PG framework that leverages cognitive hierarchy and action-conditional policies for maximizing MI among agents and maximizing agents' individual rewards. We derive an analytical lower-bond for MI estimated during InfoPG and provide mathematical reasoning underlying InfoPG's performance.

2. We propose a fully-decentralized graph-based communication and $k$-level reasoning structure to enable theory of mind for coordinating agents and maximizing their shared utility.

3. We propose a generalized variant of InfoPG and derive an MI upper-bound to modulate MI among agents depending on cooperativity of agents and environment feedback. We demonstrate the utility of this generalization in solving an instance of the Byzantine Generals Problem (BGP), in a fully decentralized setting.

4. We present quantitative results that show InfoPG sets the SOTA performance in learning emergent cooperative behaviors by converging faster and accumulating higher team rewards.

## 2 RELATED WORK

Cooperative MARL studies can be subdivided into two main lines of research, (1) learning direct communication among agents to promote coordination (Foerster et al., 2016; Das et al., 2018; Sukhbaatar et al., 2016; Kim et al., 2019) and, (2) learning to coordinate without direct communication (Foerster et al., 2017; Palmer et al., 2017; Seraj et al., 2021b). Our work can be categorized under the former. Hierarchical approaches are also prevalent for learning coordination in MARL (Seraj & Gombolay, 2020; Ghavamzadeh & Mahadevan, 2004; Amato et al., 2019; Seraj et al., 2021a). We consider MARL problems in which the task in hand is of cooperative nature and agents can directly communicate, when possible. Unlike these studies, however, we improve our interacting agents from coexisting to strategic by enabling the recursive $k$-level reasoning for decision-making.

Researchers have shown that maximizing MI among agents leads to maximizing the joint entropy of agents' decisions, which in turn, improves the overall performance in MARL (Malakar et al.,

2012; Kim et al., 2020). As such, prior work has sought to increase MI by introducing auxiliary MI regularization terms to the objective function (Kim et al., 2020; Wang et al., 2019; Jaques et al., 2019). These prior works adopt a centralized paradigm, making them less relevant to our F-Dec setting. Model of Other Agents (MOA) was proposed by Jaques et al. (2019) as a decentralized approach that seeks to locally push the MI lower-bound and promote collaboration among neighboring agents through predicting next-state actions of other agents. In all of the mentioned approaches, the amount of MI maximization objective that should be integrated into the overall policy objective is dictated through a $\beta$ regularization parameter. In our work, however, we reformulate an agent's policy to be directly conditional on the policies of its neighbors and therefore, we seek to reason about MI among agents in our PG update without ad-hoc regularization or reward shaping.

Among prior work seeking to enable $k$-level reasoning for MARL, Wen et al. (2019) presented Probabilistic Recursive Reasoning (PR2), an opponent modeling approach to decentralized MARL in which agents create a variational estimate of their opponents' level $k-1$ actions and optimize a joint Q-function to learn cooperative policies without direct communication. Wen et al. (2020) later extended the PR2 algorithm for generalized recursive depth of reasoning. In InfoPG, we establish the inherent connection between $k$-level reasoning and MI, a link that has not been explored in prior work. Moreover, we bypass the need for modeling other agents through direct communication and $k$-level action-conditional policies, and giving InfoPG agents the ability to recursively reason about their teammates' actions through received messages and with any arbitrary rationalization depths.

## 3 Preliminaries

**Problem Formulation** – We formulate our setup as a Multi-Agent Fully Dec-POMDP (MAF-Dec-POMDP), represented by an 8-tuple $\langle \{\mathcal{G}_t\}_{t \geq 0}, \mathcal{N}, \mathcal{S}, \mathcal{A}, \Omega, \{\mathcal{R}^i\}_{i \in \mathcal{N}}, \mathcal{P}, \gamma \rangle$. $\mathcal{N}$ is the set of all interacting agents in the environment in which index $i$ represents the index of an agent. $\mathcal{G}_t = \langle \mathcal{N}, \mathcal{E}_t \rangle$ is a time-varying, undirected communication graph in which agents $i, j \in \mathcal{N}$ are vertices and $\mathcal{E}_t \subseteq \{(i, j) : i, j \in \mathcal{N}, i \neq j\}$ is the edge set. The two agents $i$ and $j$ can only share information at time $t$ if $(i, j) \in \mathcal{E}_t$. State space $\mathcal{S}$ is a discrete set of joint states, $\mathcal{A}$ represents the action space, $\Omega$ is the observation space, and $\gamma \in [0, 1)$ is the temporal discount factor for each unit of time.

At each step, $t$, an agent, $i$, receives a partial observation, $o_t^i \in \Omega$, takes an action, $a_t^i \in \mathcal{A}$, and receives an immediate individual reward, $r_t^i \in \{\mathcal{R}^i\}_{i \in \mathcal{N}}$. Taking joint actions, $\bar{a}$, in the joint states, $\bar{s}$, leads to changing the joint states to $\bar{s}' \in \mathcal{S}$, according to the state transition probabilities, $\mathcal{P}(\bar{s}'|\bar{s}, \bar{a})$. Our model is *fully* decentralized since agents take actions locally and receive individual rewards for their actions according to their own reward function. Moreover, each agent is also equipped with a local optimizer to update its individual policy through its local reward feedback. Accordingly, we can reasonably assume that agents' choices of actions are conditionally independent given the current joint states (Zhang et al., 2018). In other words, if $\bar{\pi} : \mathcal{S} \times \mathcal{A} \to [0, 1]$ is the joint state-conditional policy, we assume that $\bar{\pi}(\bar{a}|\bar{s}) = \Pi_{i \in \mathcal{N}} \pi^i(a^i|\bar{s})$. Note that Dec-POMDPs are allowed to facilitate local inter-agent communication (Zhang et al., 2018; Oliehoek & Amato, 2016; Melo et al., 2011).

**Policy Gradient (PG) and Actor-Critic (AC) Methods** – The policy gradient methods target at modeling and optimizing the policy $\pi^i$ directly by parametrizing the policy, $\pi_\theta^i(a_t^i|s_t)$. Actor-Critic (AC) is a policy gradient method in which the goal is to maximize the objective by applying gradient ascent and directly adjusting the parameters of policy, $\pi_\theta^i$, through an *actor* network. The actor, updates the policy distribution in the direction suggested by a *critic*, which estimates the action-value function $Q^w(s_t, a_t^i)$ (Tesauro, 1995). By the policy gradient theorem (Sutton & Barto, 2018), the gradient by which the objective in AC, $J(\theta)$, is maximized can be shown as $\nabla_\theta J(\theta) = \mathbb{E}_{\pi_\theta^i} \left[ \nabla_\theta \log \pi_\theta^i(a_t^i|o_t^i) Q^w(\bar{s}_t, a_t^i) \right]$, where $a_t^i$ and $o_t^i$ are agent $i$'s action and observation.

**Mutual Information (MI)** – MI is a measure of the reduction in entropy of a probability distribution, $X$, given another probability distribution $Y$, where $H(X)$ denotes the entropy of $X$ and $H(X|Y)$ denotes the entropy of the conditional distribution of $X$ given $Y$ (Kraskov et al., 2004; Poole et al., 2019). By expanding the Shannon entropy of $X$ and $X|Y$, we can compute the MI as in Eq. 1.

$$I(X; Y) = H(X) - H(X|Y) = \sum_{y \in Y} p_Y(y) \sum_{x \in X} p_{X|Y=y}(x) \log \left( \frac{p_{X|Y=y}(x)}{p_X(x)} \right) \quad (1)$$

In our work, $X$ and $Y$ are distributions over actions given a specific state, for two interacting agents. In an arbitrary Markov game with two agents $i$ and $j$ and with policies $\pi_i$ and $\pi_j$, if $\pi_i$ gains MI by viewing $\pi_j$, then agent $i$ will make a more informed decision about its sampled action and vice versa.

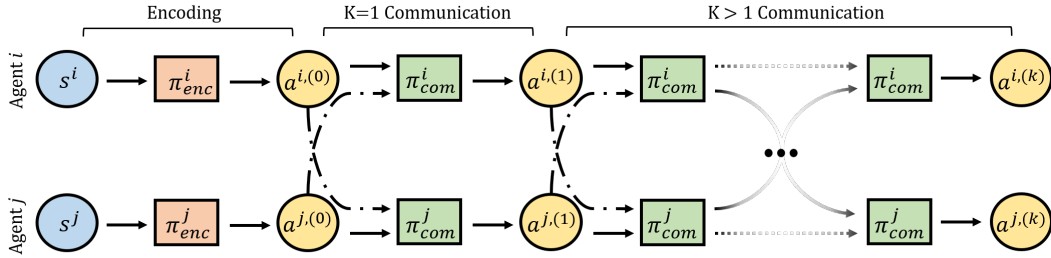

Figure 1: An instance of the information flow in a $k$-level decision hierarchy between two agents $i$ and $j$ for calculating their level-$k$ strategies. Level-zero actions (e.g., $a^{i,(0)}$) represent the naive non-strategic actions.

# 4 MUTUAL INFORMATION MAXIMIZING POLICY GRADIENT

In this section, we first present an algorithmic overview of the InfoPG framework and then introduce the InfoPG objective by covering the logistics of building an iterated $k$-level decision making strategy for agents with bounded rationality. We then explore the relation of InfoPG with MI and derive an analytical lower-bound on the MI between the action-conditional policies of interacting agents.

## 4.1 ALGORITHMIC OVERVIEW

Consider an MAF-Dec-POMDP introduced in Section 3, with $N$ agents where each agent is equipped with an *encoding* and a *communicative* policy (see Fig. 1). At the beginning of a new rollout, each agent receives a state observation from the environment and produces an initial action (i.e., a guess action) using its encoding policy. Each agent $i$ has a neighborhood of agents it can communicate with, shown with $j \in \Delta_t^i$ where $|\Delta_t^i|$ is the number of agent $i$'s physical neighbors (i.e., within close proximity). Next, depending on the level of $k$ in the decision hierarchy, agents communicate their action guesses (high-dimensional latent distributions) with their neighbors $k$ times and update their action guess iteratively using their communicative policy. The level-$k$ action is then executed by all agents and a local reward is given to each agent separately. This process continues until either the environment is solved successfully, or some maximum number of cycles has been attained. For each timestep, $t$, of the policy rollout and for each agent $i$, the gradient of the log probability is computed, scaled by the instantaneous advantage, $A_t^i$, and the encoding and communicative policies are updated. This process repeats until convergence of the cumulative discounted rewards across all agents. Please refer to Appendix, Section A.1 for pseudocode and details of our training and execution procedures.

## 4.2 DEEP REASONING: DECISION-MAKING UNDER $k$-LEVEL RATIONALIZATION

We leverage from cognitive hierarchy theory (Camerer et al., 2004) and strategic game theory (Myerson, 2013), wherein each agent has $k$-levels of conceptualization of the actions its neighbors might take under bounded rationality. $k$-level reasoning assumes that agents in strategic games are not fully rational and therefore, through hierarchies of iterated rationalizability, each agent bases its decisions on its predictions about the likely actions of other agents (Ho & Su, 2013). According to $k$-level theory, strategic agents can be categorized by the *depth* of their strategic thought (Nagel, 1995). For example, consider the two agent case shown in Fig. 1, with $k = 2$. At $k = 1$ agent $i$ makes decisions based off its own observed state, and a guess of what agent $j$ will do (Agent $i$ will assume agent $j$ is naive and non-strategic). A more sophisticated agent $i$ with a strategy of level $k = 2$ however, makes decisions based off the rationalization that agent $j$ has a level-one guess of $i$'s strategy. Inductively, this line of reasoning can be extended to any $k$. Notice that the conditioning of agent $i$'s policy on agent $j$'s perceived actions is an action-conditional distribution. In InfoPG, we give agents the ability to communicate with their latent guess-action distributions in $k$ iterated reasoning steps and rationalize their action decisions at level $k$ to best respond to their teammates' level $k - 1$ actions.

In the following, we represent the level of rationality for an agent by the superscript $(k)$ where $k \in \mathbb{N}$. Denoting $\pi^{j,(k)}$ as the level-$k$ policy of agent $j$, it can be shown that under the $k$-level reasoning, agent $i$'s policy at level $k + 1$, $\pi^{i,(k+1)}$, is precisely the best response of agent $i$ to agent $j$'s policy $\pi^{j,(k)}$ (Guan et al., 2021). In other words, $\pi^{i,(k+1)} \in \text{BestResponse}(\pi^{j,(k)})$. In theory, this process can iteratively proceed until we obtain $\pi^{i,(k+2)} = \pi^{i,(k)}$, which corresponds to reaching the equilibrium strategies. In practice, however, for scenarios with many collaborating agents, the

level of $k$ is usually set to a reasonably small number for computational efficiency; since, in order to calculate the policy $\pi^{i,(k)}$, agent $i$ must calculate not only its own policy at level $k$, but also all policies of all its neighbors for all $k \in \{1, 2, 3, \cdots, k-1\}$ at each time-step $t$.

## 4.3 InfoPG Objective

Our approach, InfoPG, equips each agent with an action-conditional policy that performs actions based on an iterated $k$-level rationalization of its immediate neighbors' actions. This process can be graphically described as presented in Figure 1, in which the information flow in a $k$-level reasoning between two agents $i$ and $j$ is shown. Note that in practice, any number of agents can be applied. Considering agent $i$ as the current agent, we represent $i$'s decision-making policy as $\pi^i_{tot} = [\pi^i_{enc}, \pi^i_{com}]$, in which $\pi^i_{enc}(a^i_t|o^i_t)$ is the state-conditional policy that maps $i$'s observed states to actions. $\pi^i_{com}(a^{i,(k)}_t|a^{i,(k-1)}_t, a^{j,(k-1)}_t)$ is the action-conditional policy that maps agent $i$'s action at level $(k-1)$ along with the actions of $i$'s neighbors (in this case agent $j$) at level $k-1$, to an action for agent $i$ in the $(k)$-th level of decision hierarchy, $a^{i,(k)}_t$. Therefore, pursuant to the general PG objective, we define the basic form of our modified information-based objective, as in Eq. 2, where $\Delta^i_t$ is the set of $i$'s immediate neighbors in communication graph at time $t$ and $G_t$ is the return.

$$\nabla^{\text{InfoPG}}_\theta J(\theta) = \mathbb{E}_{\pi^i_{tot}} \left[ G^i_t(o^i_t, a^i_t) \sum_{j \in \Delta^i_t} \nabla_\theta \log(\pi^i_{tot}(a^{i,(k)}_t|a^{i,(k-1)}_t, a^{j,(k-1)}_t, \ldots, a^{i,(0)}_t, a^{j,(0)}_t, o^i_t)) \right] \quad (2)$$

Eq. 2 describes the form of InfoPG's objective function. Depending on the use case, we can replace the return, $G^i_t$, with action-values, $Q^i_t$, shown in Eq. 3, and present the InfoPG as a Monte-Carlo PG method. We can also replace the returns with the advantage function, $A^i_t$, as shown in Eq. 4, and present the AC variant of the InfoPG objective (Sutton & Barto, 2018).

$$G^i_t(o^i_t, a^i_t) = Q^i_t(o^i_t, a^i_t) \quad \text{s.t.} \quad Q^i_t(o^i_t, a^i_t) \geq 0 \tag{3}$$

$$G^i_t(o^i_t, a^i_t) = A^i_t(o^i_t, a^i_t) = Q^i_t(o^i_t, a^i_t) - V^i_t(o_t) \tag{4}$$

Leveraging Eq. 3 and 4, we present two variants of the InfoPG objective. The first variant is the MI maximizing PG objective, which utilizes Eq. 3. The non-negative action-value condition in Eq. 3 implies non-negative rewards from the environment, a common reward paradigm utilized in prior work (Fellows et al., 2019; Liu et al., 2020; Kostrikov et al., 2018). By applying this condition, InfoPG only moves in the direction of maximizing the MI between cooperating agents (see Theorem 2). We refer to the second variant of our InfoPG objective shown in Eq. 4 as Advantage InfoPG (Adv. InfoPG), in which we relax the non-negative rewards condition. Adv. InfoPG modulates the MI among agents depending on cooperativity of agents and environment feedback (see Theorem 3).

## 4.4 Bayesian Expansion of the Policy

The action-conditional policy conditions an agent's action at the $k$-th level of the decision hierarchy on the actions of other agents at level $k-1$; however, to relate our $k$-level formulation to MI, we seek to represent an agent's action at a particular level $k$ to be dependent on the actions of other agents at same level $k$. We present Theorem 1 to introduce the gradient term in the InfoPG objective in Eq. 2 which relates level-$k$ actions of the cooperating agents in their respective decision hierarchies. Please refer to the Appendix, Section A.5 for a detailed proof of Theorem 1.

**Theorem 1.** *The gradient of the log probability's level-$k$ action-distribution in the InfoPG objective (Eq. 2) for an agent, $i$, with neighbors $j \in \Delta^i_t$ and policy $\pi^i_{tot}$ that takes the Maximum a Posteriori (MAP) action $a^{i,(k)}_t$, can be calculated iteratively for each level $k$ of rationalization via Eq. 5.*

$$\nabla \log(\pi^i_{tot}(a^{i,(k)}_t = MAP \mid a^{i,(k-1)}_t, a^{j,(k-1)}_t, \ldots, o^i_t)) = \nabla \log(\pi^i_{com}(a^{i,(k)}_t = MAP \mid a^{j,(k)}_t = MAP)) \tag{5}$$

## 4.5 InfoPG and Mutual Information Lower-Bound

Using the fact that $\nabla \log(\pi^i_{tot}(a^{i,(k)}_t|.))$ is directly proportional to $\nabla \log(\pi^i_{com}(a^{i,(k)}_t|a^{j,(k)}_t))$ from Eq. 5, we can show that the gradient of our communicative policy implicitly changes MI. Since MI is empirically hard to estimate, we instead derive a lower-bound for MI which is dependent on the action-conditional policy of an agent. We show that increasing the probability of taking an action from the action-conditional policy will increase the derived lower-bound, and consequently, the MI.

**Theorem 2.** *Assuming the actions $(a_t^{i,(k)})$ and $a_t^{j,(k)}$ to be the Maximum a Posteriori (MAP) actions, the lower-bound to MI, $I^{(k)}(i;j)$, between any pair of agents $i$ and $j$ that exist in the communication graph $\mathcal{G}_t$ can be calculated w.r.t to agent $i$'s communicative policy as shown in Eq. 6.*

$$\pi_{com}^i(a_t^{i,(k)}|a_t^{j,(k)})\log(\pi_{com}^i(a_t^{i,(k)}|a_t^{j,(k)})) \leq I^{(k)}(i;j) \tag{6}$$

**Proof –** Without loss of generality, we consider two agents $i, j \in \mathcal{N}$ with action-conditional policies $\pi^i(a^i|a^j)$ and $\pi^j(a^j|a^i)$ (note that the time, $t$, and the rationalization level, $(k)$, indices are removed for notational brevity). We refer to the marginalizations of $i$'s and $j$'s action-conditional policies as *priors* which can be denoted as $p(a^i) = \sum_{a^j \in \mathcal{A}} \pi^i(a^i|a^j)$ and $p(a^j) = \sum_{a^i \in \mathcal{A}} \pi^j(a^j|a^i)$. We assume uniformity of the priors, as done previously by Prasad (2015), such that $p(a^i) = p(a^j) = \frac{1}{|\mathcal{A}|}$, where $|\mathcal{A}|$ is the action-space dimension. For a detailed discussion on the validity of the uniformity of priors assumption, please refer to the Appendix, Section A.6. Since MI is a marginalization across all actions in the action-space, a lower-bound exists at a particular $a_{max}^i$, which is the MAP action. As such, starting from the basic definition of MI in Eq. 1, we derive:

$$I(i;j) = \sum_{a^j \in \mathcal{A}} p(a^j) \sum_{a^i \in \mathcal{A}} \pi^i(a^i|a^j) \log\left(\frac{\pi^i(a^i|a^j)}{p(a^i)}\right) = \sum_{a^j \in \mathcal{A}} \frac{1}{|\mathcal{A}|} \sum_{a^i \in \mathcal{A}} \pi^i(a^i|a^j) \log(|\mathcal{A}|\pi^i(a^i|a^j)) \tag{7}$$

$$\geq |\mathcal{A}|\pi^i(a_{max}^i|a^j)\left(\log\left(\pi_i(a_{max}^i|a^j)\right) + \log\left(|\mathcal{A}|\right)\right) \geq \pi^i(a_{max}^i|a^j)\log\left(\pi^i(a_{max}^i|a^j)\right) \quad \blacksquare \tag{8}$$

We now seek to relate the last term in Eq. 8 to the gradient term $\nabla \log(\pi_{com}^i(a_t^{i,(k)}|a_t^{j,(k)}))$ and variation of MI. By monotonicity of log, maximizing $\log(\pi^i(a_{max}^i|a^j))$ is equivalent to maximizing the $\pi^i(a_{max}^i|a^j)$ term. Therefore, according to Theorem 2 and Eq. 8, the gradient updates will raise our lower-bound, which will maximize the MI. Since the sign of environment rewards and therefore, the sign of $Q_t(o_t, a_t)$ in Eq. 3 is strictly non-negative, the gradient ascent updates will always move $\log(\pi_{com}^i(a_t^{i,(k)}|a_t^{j,(k)}))$ either up or not-at-all, which will have a proportional effect on the MI given the lower-bound. Note that we leveraged the non-negative reward condition so that the rationalization of $a_t^{i,(k)}$ given $a_t^{j,(k)}$ is only non-negatively reinforced. As such, if agent $i$'s action-conditional policy on agent $j$'s action does not obtain a positive reward from the environment, the lower-bound of MI stays constant, and thus so does MI. Conversely, if a positive reward is received by the agent, then the lower-bound of MI will strictly increase, leading to lowering the conditional entropy for taking the action that yielded the positive feedback from the environment.

## 4.6 Advantage InfoPG and the Byzantine Generals Problem

While continually maximizing MI among agents is desired for improving the degree of coordination, under some particular collaborative MARL scenarios such MI maximization may be detrimental. We specifically discuss such scenarios in the context of Byzantine Generals Problem (BGP). The BGP describes a decision-making scenario in which involved agents must achieve an optimal collaborative strategy, but where at least one agent is corrupt and disseminates false information or is otherwise unreliable (Lamport et al., 2019). BGP scenarios are highly applicable to cooperative MARL problems where there exists an untrainable fraudulent agent with a bad policy (e.g., random) in the team. Coordinating actions with such a fraudulent agent in a collaborative MARL setting can be detrimental. We note that BGPs are particularly challenging to solve in the fully decentralized settings (Peng et al., 2021; Allen-Zhu et al., 2020).

Here, we elaborate on our Adv. InfoPG variant introduced in Eq. 4 and show its utility for intelligently modulating MI depending on the cooperativity of agents. Intuitively, the advantage function evaluates how much better it is to take a specific action compared to the average, general action at the given state. Without the non-negative reward condition in InfoPG, the Adv. InfoPG objective in Eq. 4 does not *always* maximize the MI locally, but, instead, benefits from both positive and negative experiences as measured by the advantage function to increase the MI in the long run. Although instantaneous experiences may result in a negative advantage, $A_t^i$, and reducing the MI lower-bound in Eq. 6, we show that Adv. InfoPG in fact *regularizes* an MI upper-bound, making it suitable for BGP scenarios while also having the benefit of learning from larger number of samples. In Adv. InfoPG we equip each agent with an action-conditional policy and show that under $k$-level reasoning, the tight bounds of MI between agents is regularized (shifted up and down) depending on the sign of the received advantage. We note that despite this local MI regularization, in the long run we expect the MI bounds to increase since policy gradient seeks to maximize local advantages during gradient ascent.

**Upper-Bound of MI** – Here, we derive an MI upper-bound dependent on the action-conditional policy of an agent and show that the gradient updates in Adv. InfoPG have a proportional effect on this upper-bound. We show that under $k$-level rationalization, the tight bound of MI between agents is regularized (shifted up or down) depending on the sign of the received advantage value, $A_t^i$.

**Theorem 3.** *Assuming the same preliminaries as in Theorem 2, the upper-bound to MI, $I^{(k)}(i;j)$, between agents $i$ and $j$ w.r.t agent $i$'s level-$k$ action-conditional policy can be calculated as in Eq. 9.*

$$I^{(k)}(i;j) \leq 2\log(|\mathcal{A}|) + 2\log(\pi_{com}^i(a_t^{i,(k)}|a_t^{j,(k)})) \tag{9}$$

**Proof** – We start from the definition of conditional entropy. The conditional entropy is an expectation across all $a^i$ and considering the fact that the $-\log(.)$ is a convex function, Jensen's inequality (Ruel & Ayres, 1999) can be applied to establish an upper-bound on conditional entropy. We derive:

$$H(\pi^i|\pi^j) = -\sum_{a^i \in \mathcal{A}} \pi^i(a^i|a^j)\log(\pi^i(a^i|a^j)) = \sum_{a^i \in \mathcal{A}} \pi^i(a^i|a^j)(-\log(\pi^i(a^i|a^j))) \tag{10}$$

$$\xrightarrow{\text{Jensen's inequality}} H(\pi^i|\pi^j) \geq -\log(\sum_{a^i \in \mathcal{A}} \pi^i(a^i|a^j)^2) \tag{11}$$

Now, we leverage the basic MI definition in Eq. 1. We note that $H(p(a^i))$ has a constant value of $\log(|\mathcal{A}|)$ given the uniform prior assumption. Accordingly, plugging in the bound in Eq. 11 and evaluating at the MAP action results in an upper-bound for MI, as shown below.

$$I(i;j) = H(p(a^i)) - H(\pi^i|\pi^j) = -H(\pi^i|\pi^j) + \log(|\mathcal{A}|) \leq \log(\sum_{a^i \in \mathcal{A}} \pi^i(a^i|a^j)^2) + \log(|\mathcal{A}|) \tag{12}$$

$$\leq \log\left(|\mathcal{A}|\pi^i(a_{max}^i|a^j)^2\right) + \log(|A|) \leq 2\log(|\mathcal{A}|) + 2\log\left(\pi^i(a_{max}^i|a^j)\right) \quad \blacksquare \tag{13}$$

Considering the Adv. InfoPG objective in Eq. 4, depending on the sign of $A_t^i$, the gradient ascent either increases or decreases $\log(\pi_{com}^i(a_t^{i,(k)}|a_t^{j,(k)})$, which will have a proportional regulatory effect on the MI given our bounds in Eq. 6 and 9. Specifically, when agent $i$ receives negative advantage from the environment, it means agent $i$'s reasoning of $a_t^{i,(k)}$ given $a_t^{j,(k)}$, resulted in a negative outcome. Therefore, to reduce agent $j$'s negative influence, our gradient updates in Eq. 4 will decrease the MI upper-bound between $i$ and $j$ so that the conditional entropy at level $k$ is increased. This bears similarity to Soft actor-critic (Haarnoja et al., 2018), where regularization with entropy allows agents to *explore* more actions. As such, during Adv. InfoPG updates, the MI constraint becomes adaptive to instantaneous advantage feedback. Such property can be effective in BGP scenarios to reduce the negative effect of misleading information received from a fraudulent agent.

## 5 EMPIRICAL EVALUATION

**Evaluation Environments** – We empirically validate the utility of InfoPG against several baselines in four cooperative MARL domains that require high degrees of coordination and learning collaborative behaviors. Our testing environments include: (1) Cooperative Pong (Co-op Pong) (Terry et al., 2020), (2) Pistonball (Terry et al., 2020), (3) Multiwalker (Gupta et al., 2017; Terry et al., 2020) and, (4) StarCraft II (Vinyals et al., 2017), i.e., the 3M (three marines vs. three marines) challenge. We modified the reward scheme in all four domains to be individualistic such that agents only receive a local reward feedback as per our MAF-Dec-POMDP formulation in Section 3. For environment descriptions and details, please refer to the Appendix, Section A.8.

**Baselines** – We benchmark our approach (both InfoPG in Eq. 3 and Adv. InfoPG in Eq. 4) against four *fully-decentralized* baselines: (1) Non-communicative A2C (NC-A2C) (Sutton & Barto, 2018), (2) Consensus Update (CU) (Zhang et al., 2018), (3) Model of Agents (MOA) (Jaques et al., 2019) and, (4) Probabilistic Recursive Reasoning (PR2) (Wen et al., 2019). In the NC-A2C, each agent is controlled via an individualized actor-critic network without communication. The CU approach shares the graph-based communication among agents with InfoPG but lacks the $k$-level reasoning in InfoPG's architecture. Thus, the CU baseline is communicative and non-rational. MOA, proposed by Jaques et al. (2019), is a decentralized cooperative MARL method in which agents benefit from action-conditional policies and an MI regularizer dependent on the KL-Divergence between an agent's prediction of its neighbors' actions and their true actions. PR2, proposed by Wen et al. (2019) is an opponent modeling approach to decentralized MARL in which agents create a variational estimate of their opponents' level $k-1$ actions and optimize a joint Q-function to learn cooperative policies through $k$-level reasoning without direct communication.

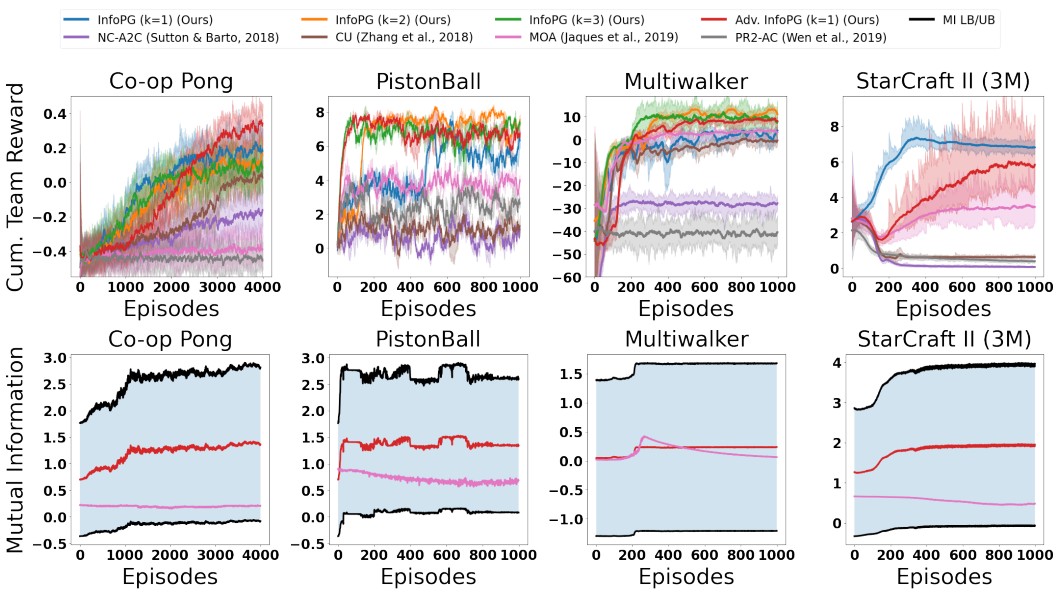

Figure 2: (Top Row) Team rewards obtained across episodes as training proceeds. The shaded regions represent standard error. Our Adv. InfoPG continually outperforms all baselines across all domains and in both training and testing (see Table 1). (Bottom Row) The MI ablation study results, comparing the MI variations between InfoPG and MOA (MI-based baseline) where InfoPG demonstrates a higher final average MI estimate across all domains. The shaded blue region represents the area between InfoPG's lower and upper bounds on MI.

## 6 Results, Ablation Studies, and Discussion

In this section, we assess the performance and efficiency of our frameworks in the four introduced domains and against several recent, fully-decentralized cooperative MARL approaches. Following an analysis of performance under $k$-level reasoning and an MI ablation study, we present a case-study, namely the *fraudulent agent experiment*, to investigate the utility of our MI-regularizing InfoPG variant (i.e. Adv. InfoPG in Eq. 4) against MI-maximizing baselines, such as MOA (Jaques et al., 2019) and InfoPG, in BGP scenarios. We provide further ablation studies, such as a level-$k$ policy interpretation for InfoPG and a scalability analysis in the Appendix, Section A.7.

**Baseline Comparison** – The top row in Fig. 2 depicts the team rewards obtained by each method across episodes as training proceeds. In all four domains, both InfoPG and Adv. InfoPG demonstrate sample-efficiency by converging faster than the baselines and achieving higher cumulative rewards. Table 1 presents the mean (±standard error) cumulative team rewards and steps taken by agents to win the game by each method at convergence. Table 1 shows that InfoPG and Adv. InfoPG set the state-of-the-art for learning challenging emergent cooperative behaviors in both discrete and continuous domains. Note that, while in Pistonball fewer number of taken steps means better performance, in Co-op Pong and Multiwalker more steps shows a superior performance.

**Mutual Information Variation Analysis** – The bottom row in Fig. 2 shows our MI study results comparing the MI variations between InfoPG and MOA (the MI-based baseline). The MI for InfoPG is estimated as the average between lower and upper bounds defined in Eqs. 6 and 9. As depicted, InfoPG demonstrates a higher final average MI estimate across all domains. Note the concurrency of InfoPG's increase in MI estimates (bottom row) and agents' performance improvements (Fig. 2, top row). This concurrency supports our claim that our proposed policy gradient, InfoPG, increases MI among agents which results in learning emergent collaborative policies and behavior.

**Deep Reasoning for Decision Making: Evaluating $k$-Level InfoPG** – We evaluate the performance of our method for deeper levels of reasoning $k \in \{2, 3\}$ in Co-op Pong, Pistonball and Multiwalker domains. The training results are presented in Fig. 2. As discussed in Section 4.2, in the smaller domain with fewer collaborating agents, the Co-op Pong, agents reach the equilibrium cooperation strategy (i.e., $\pi^{i,(k+2)} = \pi^{i,(k)}$) even with one step of reasoning, and increasing the level of $k$ does not significantly change the performance. However, in the more complex domains with more agents, Pistonball and Multiwalker, as the level of rationalization goes deeper in InfoPG (i.e., $k = 2$ and $k = 3$), agents can coordinate their actions better and improve the overall performance.

Table 1: Reported results are Mean (Standard Error) from 100 testing trials. For all tests, the final training policy at convergence is used for each method and for InfoPG and Adv. InfoPG, the best level of $k$ is chosen.

| Domain | InfoPG | | Adv. InfoPG | | MOA | | CU | | NC-A2C | | PR2-AC | |
|---|---|---|---|---|---|---|---|---|---|---|---|---|
| | $\mathcal{R}$ | #Steps | $\mathcal{R}$ | #Steps | $\mathcal{R}$ | #Steps | $\mathcal{R}$ | #Steps | $\mathcal{R}$ | #Steps | $\mathcal{R}$ | #Steps |
| Co-op Pong | **0.127** | **202.9** | **0.25** | **212.9** | -0.3 | 58.2 | 0.13 | 152.0 | 0.04 | 102.925 | -0.84 | 36.8 |
| | **(0.00)** | **(1.35)** | **(0.00)** | **(1.24)** | (0.00) | (0.32) | (0.00) | (0.96) | (0.00) | (0.93) | (0.00) | (0.34) |
| Pistonball | **7.47** | **17.44** | **7.33** | **28.31** | 4.10 | 91.66 | 1.06 | 136.3 | 0.89 | 138.3 | 1.90 | 140.4 |
| | **(0.02)** | **(0.29)** | **(0.02)** | **(0.43)** | (0.03) | (0.76) | (0.04) | (0.83) | (0.04) | (0.83) | (0.02) | (0.68) |
| Multiwalker | **3.56** | **474.2** | **11.81** | **500.0** | 0.66 | 489.3 | -1.7 | 490.2 | -66 | 80.75 | -84 | 94.17 |
| | **(0.20)** | **(1.39)** | **(0.11)** | **(0.80)** | (0.15) | (1.09) | (0.26) | (1.30) | (0.18) | (0.20) | (0.37) | (1.40) |
| StarCraft II | **4.40** | **30.79** | **3.73** | **44.5** | 2.78 | 27.7 | 0.24 | 58.72 | 0.00 | 60.0 | 0.64 | 27.4 |
| | **(0.01)** | **(0.05)** | **(0.02)** | **(0.13)** | (0.02) | (0.07) | (0.00) | (0.06) | (0.00) | (0.00) | (0.00) | (0.08) |

**The Fraudulent Agent Experiment: Regularising MI** – To assess our Adv. InfoPG's (Eq. 4) regulatory effect based on agents' cooperativity, we perform an experiment, demonstrated in Fig. 3a, which is performed in the Pistonball domain and is intended to simulate an instance of the BGP in which the middle piston is equipped with a fully random policy throughout the training (i.e., the Byzantine piston is not controllable by any InfoPG agent). Maximizing the MI with this fraudulent agent is clearly not desirable and doing such will deteriorate the learning performance.

Fig. 3b presents the fraudulent agent experiment training results for Adv. InfoPG, InfoPG and MOA. As shown and comparing with Fig. 2, existence of a fraudulent agent significantly deteriorates the learning performance in MOA and InfoPG as these approaches always seek to maximize the MI among agents. This is while InfoPG still outperforms MOA since by only using strictly non-negative rewards, the coordination in InfoPG is only positively reinforced, meaning that InfoPG only increases MI when the reward feedback is positive. Adv. InfoPG shows the most robust performance compared to InfoPG and MOA. Adv. InfoPG modulates the MI depending on the observed short-term coordination performance. As discussed in Section 4.6, if the advantage is negative, the gradient ascent in Adv. InfoPG will decreases the MI upper-bound between agents, leading to increasing the conditional entropy and taking more exploratory (i.e., less coordinated) actions.

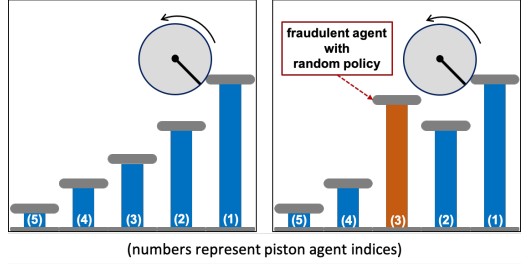 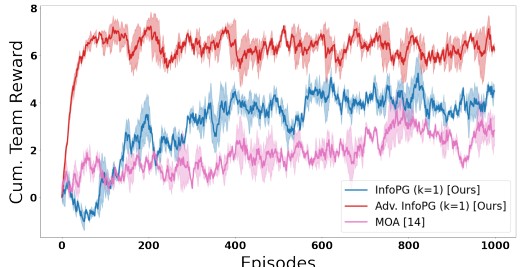

(a) The Fraudulent Agent Experiment: Scenario      (b) The Fraudulent Agent Experiment: Result

Figure 3: The fraudulent agent experiment scenario (Fig. 3a) and training results (Fig. 3b) in the Pistonball domain, comparing the team reward performance for Adv. InfoPG (Eq. 4), InfoPG (Eq.3) and MOA.

**Limitations and Future Work** – Useful MI between agents becomes hard to capture in cases, such as Co-op Pong domain, where an agent's action influences its neighbors with some delay. Moreover, MI maximization by applying the strictly non-negative reward condition in InfoPG objective (Eq. 3) comes at the cost of zeroing out negative experiences which may have an impact on sample-efficiency.

## 7 CONCLUSION

We leverage iterated $k$-level reasoning from cognitive hierarchy theory and present a collaborative, fully-decentralized MARL framework which explicitly maximizes MI among cooperating agents by equipping each agent with an action-conditional policy and facilitating iterated inter-agent communication for hierarchical rationalizability of action-decisions. We analytically show that the design of our MI-based PG method, increases an MI lower-bound, which coincides with improved cooperativity among agents. We empirically show InfoPG's superior performance against various baselines in learning cooperative policies. Finally, we demonstrate that InfoPG's regulatory effect on MI makes it Byzantine-resilient and capable of solving BGPs in fully-decentralized settings.

ACKNOWLEDGMENTS

This work was sponsored by the Office of Naval Research (ONR) under grant N00014-19-1-2076 and the Naval Research Lab (NRL) under the grant N00173-20-1-G009.

ETHICAL STATEMENT

Our experiments show that theory of mind and cognitive hierarchy theory can be enabled for teams of cooperating robots to assist strategic reasoning. We note that our research could be applied for good or otherwise, particularly in an adversarial setting. Nonetheless, we believe democratizing this knowledge is for the benefit of society.

REPRODUCIBILITY STATEMENT

We have taken several steps to ensure the reproducibility of our empirical results: we submitted, as a supplementary material, our source code for training and testing InfoPG and all the four baselines in all four domains. The source code is accompanied with a README with detailed instructions for running each file. We also publicized our source code in a public repository, available online at https://github.com/CORE-Robotics-Lab/InfoPG. Additionally, we have provided the details of our implementations for training and execution as well as the full hyperparameter lists for all methods, baselines, and experiments in the Appendix, Section A.9. For our theoretical results, we provide explanations of all assumptions made, including the uniformity of priors assumption in Appendix, Section A.6, and the assumptions made for convergence proof in Appendix, Section A.4. Finally, a complete proof of our introduced Theorems 2 and 3, as well as InfoPG's convergence proof and the full proof of Theorem 1, are provided in Sections 4.5-4.6 and, Appendix, Sections A.4-A.5 respectively.

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

# A APPENDIX

## A.1 INFOPG PSEUDOCODE

Here, we provide a pseudocode to train our MI maximizing PG algorithm, InfoPG, in Algorithm 1.

---

**Algorithm 1:** Training the Mutual Information Maximizing Policy Gradient (InfoPG).

1: **Input:** Number of agents, $\mathcal{N}$, $max\_cycles$ and agents' level of iterated rationalizability, $K$
2: **Initialize:** For all agents $\{\pi_{tot}^1, \pi_{tot}^2, \cdots, \pi_{tot}^{\mathcal{N}}\}$ and $\{V^1, V^2, \cdots, V^{\mathcal{N}}\}$
3: **while** not converged **do**
4:   **for** $t = 1$ to $max\_cycles$ **do**
5:     Reset environment and receive initial observation set: $\{o_t^1, o_t^2, ..., o_t^{\mathcal{N}}\}$
6:     **for** $i = 1$ to $\mathcal{N}$ **do**
7:       Sample initial "guess" action: $a_t^{i,(0)} \sim \pi_{enc}^i(* \mid o_t^i)$, where $\pi_{enc}^i \in \pi_{tot}^i$
8:     **end for**
9:     **for** $k = 1$ to $K$ **do**
10:       **for** $i = 1$ to $\mathcal{N}$ **do**
11:         Identify neighbors by obtaining neighbor list, $j \in \Delta_t^i$, using the adjacency graph, $\mathcal{G}_t$
12:         Sample MAP: $a_t^{i,(k)} \sim \pi_{com}^i(* \mid a_t^{i,(k-1)}, \{a_t^{j,(k-1)} \mid j \in \Delta_t^i\})$, where $\pi_{com}^i \in \pi_{tot}^i$
13:       **end for**
14:     **end for**
15:     Step through environment using $\{a_t^{1,(k)}, a_t^{2,(k)}, \cdots, a_t^{\mathcal{N},(k)}\}$, and receive next states and rewards: $\{o_{t+1}^1, o_{t+1}^2, ..., o_{t+1}^{\mathcal{N}}\}$, $\{r_t^1, r_t^2, ..., r_t^{\mathcal{N}}\}$
16:     **for** $i = 1$ to $\mathcal{N}$ **do**
17:       $A_t^i = r_t^i + V^i(o_{t+1}^i) - V^i(o_t^i)$
18:       $\nabla \pi_{tot}^i = \text{ReLU}(A_t^i) \nabla \log(\pi_{tot}^i(a_t^{i,(k)} \mid \cdots))$   % For Adv. InfoPG remove the ReLU
19:       $\nabla V^i = (A_t^i)^2$
20:       Update: $\pi_{tot}^i = \pi_{tot}^i + \eta \nabla \pi_{tot}^i$
21:       Update: $V^i = V^i + \eta \nabla V^i$
22:     **end for**
23:     $\{o_t^1, o_t^2, ..., o_t^{\mathcal{N}}\} = \{o_{t+1}^1, o_{t+1}^2, ..., o_{t+1}^{\mathcal{N}}\}$
24:   **end for**
25: **end while**

---

Consider a MAF-Dec-POMDP introduced in Section 3, with $N$ agents where each agent is equipped with an *encoding* and a *communicative* policy ($\pi_{enc}$ and $\pi_{com}$, respectively), such that $\pi_{tot} = [\pi_{enc}, \pi_{com}]$ (lines 1-2). At the beginning of a new rollout, and for each timestep, $t$, within the allowed maximum number of steps, $max\_cycles$, each agent $i$ receives a state observation $o_t^i$ from the environment and produces an initial "guess" action, $a_t^{i,(0)}$ using its encoding policy (lines 5-8). Each agent $i$ has a neighborhood of agents it can communicate with, shown with $j \in \Delta_t^i$ where $|\Delta_t^i|$ is the number of agent $i$'s physical neighbors (i.e., within close proximity). Accordingly, agents obtain the list of their neighbors by using the adjacency graph, $\mathcal{G}_t$ (line 11).

Next, depending on the specified level of iterated rationalizability in the decision hierarchy, $K$, agents communicate their action guesses as higher-dimensional latent vector distributions with their neighbors $K$ times and update their action guess iteratively using their communicative policy (line 9-14). The level-$k$ action is then executed by all agents and a local reward is given to each agent separately (line 15). This process continues until either the environment is solved successfully, or the allowed maximum number of steps, $max\_cycles$, has been attained. For each timestep, $t$, of the policy rollout and for each agent $i$, first the advantage value, $A_t^i$, is computed using the critic network (line 17) and then, the gradient of the log probability is computed and scaled by the instantaneous advantage (line 18). Note that the ReLU function in line 18 is proposed to enforce the non-negative reward feedback condition in InfoPG, Eq. 3. However, other mechanisms could achieve the same effect (e.g., shaping the reward function to only include positive values). Line 19 is showing the gradient of our value estimate. The loss for our value estimate $V(s_t)$ is the cumulative discounted rewards subtracted by the true value of $s_t$, which is the advantage (the TD error). Therefore, the

gradient in line 19 is the sum of squared individual advantages. Next, the encoding and communicative policies are updated (line 20) and eventually, the critic network is updated (line 21). This process repeats until convergence of the cumulative discounted rewards across all agents. We provide our code at https://github.com/CORE-Robotics-Lab/InfoPG

## A.2 INFOPG FOR CONTINUOUS ACTION-SPACES

To deploy InfoPG in continuous action-spaces, we follow common practice for continuous action-space actor-critic methods. Continuous policies are presented as probability distributions with floating values in a certain range (e.g., $(-1, 1)$). In this case, in order to facilitate exploration we will sample agents' actions from a normal probability distribution. As such, in continuous action-spaces, the policy network (Actor) will normally have two output heads, instead of one. The two outputs of the policy network are $\mu$ and $\sigma$, the mean and standard deviation (STD) of the probability distribution, respectively. The sampled actions will be centered around the $\mu$ and the $\sigma$ determines on average, how far from the center the sample values will be. As the network gets more certain, the $\sigma$ gets smaller, meaning that we tend to exploit good actions rather than exploring.

In discrete action-spaces the loss-function was based on the log-probability (Eq. 3). In continuous action-spaces, the log-probability of a normal distribution is used, as shown in Eq. 14.

$$log_{\pi_\theta}(a|s) = -\frac{(x - \mu)^2}{2\sigma^2} - log\sqrt{2\pi\sigma^2} \tag{14}$$

In Eq. 14, the first and second term are the negative log-probability and the entropy bonus, respectively. By substituting the log-probability of the normal distribution in Eq. 14 back into the original InfoPG objective in Eq. 2, we can directly deploy the InfoPG objectives in continuous domains, such as the introduced Multiwalker (Section 5). Note that, in practice and for simplicity, the policy network can only output the mean value, $\mu$, while the STD value is fixed to a reasonable constant. Finally, to compute the MAP action for a continuous action-space (Line 12 in Algorithm 1) we note that, theoretically, the MAP action for a continuous space is just the mean action without any standard deviation from the normal distribution.

## A.3 INFOPG OBJECTIVE FUNCTION DERIVATION (EQ. 2)

The InfoPG definition as presented in Eq. 2 consists of a summation across all agents within the communication range. This equation is a simplification from the original form which uses the assumption of independence between each agents' $(k-1)$-level action probability distributions. To arrive at the InfoPG objective function in Eq. 2, we start from Eq. 15 and convert the log probability of the conditional distributions across all neighbors to a summation of log probability across all neighbors. The process can be shown as in Eq. 15-17.

$$\nabla_\theta^{\text{InfoPG}} J(\theta) = \mathbb{E}_{\pi_{tot}^i} \left[ G_t^i(o_t^i, a_t^i) \nabla_\theta \log(\pi_{tot}^i(a_t^{i,(k)} | \forall_{0 \to k} \forall_{j \in \Delta_t^i} [a_t^{i,(k-1)}, a_t^{j,(k-1)}], o_t^i)) \right] \tag{15}$$

$$= \mathbb{E}_{\pi_{tot}^i} \left[ G_t^i(o_t^i, a_t^i) \nabla_\theta \log(\prod_{0 \to k} \prod_{j \in \Delta_t^i} \pi_{tot}^i(a_t^{i,(k)} | a_t^{i,(k-1)}, a_t^{j,(k-1)}, o_t^i)) \right] \tag{16}$$

$$= \mathbb{E}_{\pi_{tot}^i} \left[ G_t^i(o_t^i, a_t^i) \sum_{j \in \Delta_t^i} \nabla_\theta \log(\pi_{tot}^i(a_t^{i,(k)} | a_t^{i,(k-1)}, a_t^{j,(k-1)}, \ldots, a_t^{i,(0)}, a_t^{j,(0)}, o_t^i)) \right] \tag{17}$$

## A.4 CONVERGENCE PROOF SKETCH FOR EQ. 2

Following the approach in prior work (Bhatnagar et al., 2009; Zhang et al., 2018), we present a convergence proof sketch for InfoPG through the Two-Time-Scale (TTS) stochastic approximation method, proposed by Borkar (1997). We note that the convergence proof for InfoPG closely follows the general Policy Gradients (PG) convergence approach presented in Bhatnagar et al. (2009) and Zhang et al. (2018), and we therefore only focus on presenting the core idea underlying convergence of InfoPG with $k$-level rationalizability. Herein, we state that since InfoPG and Consensus Update (CU) share the graph-based local communication and the *fully*-decentralized actor-critic training paradigm, all assumptions made by Zhang et al. (2018) also apply to our work and therefore, we

directly adopt the set of assumptions (i.e., specifically, assumptions 4.1 – 4.4) presented in Zhang et al. (2018) without restating them.

For an actor-critic algorithm, e.g. InfoPG, showing convergence of both the actor and critic simultaneously is challenging because the noisy performance of the actor can affect the gradient on the critic and vice versa. As such, we leverage the TTS approach, which states that in PG methods, the *actor* learns at a slower rate than the *critic* (Borkar, 1997). Therefore, according to TTS, we can show the convergence of InfoPG in two steps: (1) first, we fix the policy and analyze the convergence of the critic and, (2) with a converged critic, we analyze the convergence of the actor.

**Step 1: Critic Convergence –** We use bar notation in the following to denote vectorized quantities across agents in the environment such that, $\bar{\pi}_\theta = [\pi_\theta^1, \cdots, \pi_\theta^N]$, where $N$ is the number of agents. Moreover, a level-$k$ policy in InfoPG includes two parts: a state-conditional policy at level $k = 0$ and an action-conditional policy for higher levels of $k$. InfoPG first applies the level zero state-conditional policy to get the initial "guess" action and then recursively applies the action-conditional policy $k$ times to recursively improve the level-$k$ action-decision. We can show this process as $\pi_\theta^i = \pi_\theta^{i,(k \geq 0)}\left(\pi_\theta^{i,(k=0)}(s_t^i)\right)$. PG seeks to maximize the objective function $\bar{J}(\theta)$, shown in Eq. 18, where $\bar{d}_\pi(\bar{s}_t)$ denotes the stationary distribution of the MDP and $\bar{R}$ denotes the joint reward function, including local rewards for each agent. Additionally, we denote the transition probability of the MDP as $\bar{p}(\bar{s}_{t+1}, \bar{r}_{t+1}|\bar{s}_t, \bar{a}_t)$. Since the formulation of PG objective in Eq. 18 is biased, it is commonly replaced with the unbiased estimate of the rewards, or $\bar{Q}(\bar{s}_t, \bar{a}_t) - \bar{V}(\bar{s}_t, \bar{a}_t)$, as shown in Eq. 19

$$\bar{J}(\theta) = \sum_{\bar{s}_t \in \bar{S}} \bar{d}_\pi(s) \sum_{\bar{a}_t \in \bar{A}} \bar{\pi}_\theta(\bar{a}_t|\bar{s}_t) * \bar{R}(\bar{s}_t, \bar{a}_t) \tag{18}$$

$$= \sum_{\bar{s}_t \in \bar{S}} \bar{d}_\pi(s) \sum_{\bar{a}_t \in \bar{A}} \bar{\pi}_\theta(\bar{a}_t|\bar{s}_t) * (\bar{Q}(\bar{s}_t, \bar{a}_t) - \bar{V}(\bar{s}_t)) \tag{19}$$

The unbiased estimator in Eq. 19 can be replaced with the state-value function by using the recursive definition of the action-value function (Sutton & Barto, 2018). This substitution results in a form known as the TD-error (Sutton, 1985), where the bracketed term in Eq. 21 is the TD-error.

$$\bar{Q}(\bar{s}_t, \bar{a}_t) = \mathop{\mathbb{E}}_{\bar{s}_t \sim \bar{d}_\pi \bar{a}_t \sim \bar{\pi}_\theta} \left[\bar{R}(\bar{s}_t, \bar{a}_t) + \gamma \bar{V}(\bar{s}_{t+1})\right] \tag{20}$$

$$\bar{J}(\theta) = \sum_{\bar{s}_t \in \bar{S}} \bar{d}_\pi(s) \sum_{\bar{a}_t \in \bar{A}} \bar{\pi}_\theta(\bar{a}_t|\bar{s}_t) * \left[\bar{R}(\bar{s}_t, \bar{a}_t) + \gamma \sum_{\bar{s}_{t+1} \in \bar{S}} \bar{p}(\bar{s}_{t+1}|\bar{s}_t, \bar{a}_t)\bar{V}(\bar{s}_{t+1}) - \bar{V}(\bar{s}_t)\right] \tag{21}$$

Next, following the prior work (Bhatnagar et al., 2009; Zhang et al., 2018), we assume linear function approximation since the TD-learning-based policy evaluation may fail to converge with nonlinear function approximation (Tsitsiklis & Van Roy, 1997; Zhang et al., 2018). We note that the value function is a mapping of states (of some dimensionality) to $\mathcal{R}$. Therefore, we define $\bar{V}(\bar{s}_t) = \bar{\omega}^T \phi(\bar{s}_t)$, where $\bar{\omega}$ is a one dimensional weight vector and $\phi(\bar{s}_t)$ is a feature map that transforms the state vector to $\mathcal{R}^K$: $\phi(\bar{s}_t) = [\phi_1(\bar{s}_t), \cdots, \phi_K(\bar{s}_t)]$. Now, following Bhatnagar et al. (2009), we define the Ordinary Differential Equation (ODE) associated with the recursive update of $\bar{\omega}$ via Eq. 22, which then can be simplified to Eq. 23 using a matrix notation described in the following.

$$\dot{\omega} = \sum_{\bar{s}_t \in \bar{S}} \bar{d}_\pi(s) \sum_{\bar{a}_t \in \bar{A}} \bar{\pi}_\theta(\bar{a}_t|\bar{s}_t) * \left[\bar{R}(\bar{s}_t, \bar{a}_t) + \gamma \sum_{\bar{s}_{t+1} \in \bar{S}} \bar{p}(\bar{s}_{t+1}|\bar{s}_t, \bar{a}_t)\bar{\omega}^T \phi(\bar{s}_t + 1) - \bar{\omega}^T \phi(\bar{s}_t)\right] \tag{22}$$

$$\dot{\omega} = \Phi^T D[T(\Phi\bar{\omega}) - \Phi(\bar{\omega})] \tag{23}$$

Finding the asymptotic equilibrium of the critic, $\omega$ is equivalent to solving the above equation, when $\dot{\omega} = 0$, which is simplified when switching to matrix vector notation described below:

1. $D \in \mathbb{R}^{|\bar{S}| \times |\bar{S}|}$ is a diagonal matrix with $\bar{d}_\pi(\bar{s}_t)$ for all $\bar{s}_t \in \bar{S}$ as its elements.

2. $P \in \mathbb{R}^{|\bar{S}| \times |\bar{S}||\bar{A}|}$ is the probability matrix where $\bar{p}(\bar{s}_{t+1}|\bar{s}_t, \bar{a}_t)\bar{\pi}_\theta(\bar{s}_t|\bar{a}_t)$ represents an individual element.

3. $\Phi \in \mathbb{R}^{|\bar{S}| \times \bar{K}}$ is the feature map whose rows are $\phi(\bar{s}_t)$ for all $\bar{s}_t \in \bar{S}$.

4. $R \in \mathbb{R}^{|\bar{S}| \times |\bar{A}|}$ is a matrix where $\bar{R}(\bar{s}_t, \bar{a}_t)$ represents an individual element.

5. $\Omega \in \mathbb{R}^{|\bar{K}| \times |\bar{K}|}$ is a diagonal matrix with discount factor $\gamma$ as its elements.

6. $T : \mathbb{R}^N \to \mathbb{R}^N$ is an operator which is a mapping of the form: $T(\bar{\omega}) = R + P\Omega\bar{\omega}$.

Next, following Zhang et al. (2018), we make two assumptions that apply to InfoPG and are essential to the rest of the convergence proof presented in the following.

**Assumption 1 –** The update of the policy parameter $\theta$ includes a local projection operator, $\Gamma : \mathbb{R}^N \to \chi \subset \mathbb{R}^N$, that projects any $\theta$ onto the compact set $\chi$. Also, we assume that $\chi$ is large enough to include at least one local minimum of $\bar{J}(\theta)$.

**Assumption 2 –** The instantaneous reward $r_t^i$ is uniformly bounded for any $i \in \mathcal{N}$ and $t \geq 0$. We note that the reward boundedness assumption is rather mild and is in accordance with prior work (Zhang et al., 2018).

In analogous matrix-vector notation, and under the aforementioned assumptions made above and by Zhang et al. (2018), for a static policy in the TTS convergence setting, the $\lim_{t \to \infty} \bar{\omega} = \bar{\omega}^\star$ almost surely, where $\bar{\omega}^\star$ satisfies the equilibrium constraint $\dot{\bar{\omega}} = 0$ shown below. We note the solution seen below satisfies a similar convergence equation as seen in Bhatnagar et al. (2009).

$$\dot{\bar{\omega}} = \Phi^T D[T(\Phi\bar{\omega}^\star) - \Phi(\bar{\omega}^\star)] = 0 \tag{24}$$

$$= \Phi^T D T(\Phi\bar{\omega}^\star) = \Phi^T D \Phi(\bar{\omega}^\star) \tag{25}$$

The above ODE explains the rate of change of the critic, and when the derivative reaches zero, the critic has reached a stable equilibrium, and therefore, has converged.

**Step 2: Actor Convergence –** According to the TTS (Borkar, 1997), for the actor step, we assume a fixed, converged critic and show a stabilized equilibrium of the policy. We assume there exists an operator, $\Gamma$, which projects any vector $x \in \mathcal{R}^N \to \chi \subset \mathcal{R}^N$, where $\chi$ represents a compact set bounded by a simplex in $\mathcal{R}^N$. The use of this projection is critical to the convergence of stochastic TTS algorithms as stated by Bhatnagar et al. (2009); Zhang et al. (2018), since policies that exist outside of the set can cause unstable equilibrium. Empirically, we apply the compactness of the set of policy gradients by defining a maximum gradient norm, as stated in A.8. In Eq. 26, we define the $\Gamma$ operator for the vector field $x(.) \in \theta$, which is assumed to be a continuous function.

$$\hat{\Gamma}(x(y)) = \lim_{0 \leq \eta \to 0} \frac{\Gamma(y + \eta x(y)) - y}{\eta} \tag{26}$$

If the above limit does not converge to a singular value, we state $\hat{\Gamma}(x(y))$ results in a *set* of convergent points. With this notation we state the ODE of the policy, after being projected onto a compact set, and note that given the assumptions made above and by Zhang et al. (2018), PG almost surely moves $\bar{\theta}$ to an asymptotically stable equilibrium that satisfies the below Eq. 27. This proof analogy closely follows the single-agent convergence proofs presented in Bhatnagar et al. (2009) and Tsitsiklis & Van Roy (1997). Nevertheless, in our work, convergence to a stationary point for all agents is the goal. While the TTS approach assumes the critic to have converged, the critic does not need to be a perfect estimator. With small approximation error, Bhatnagar et al. (2009) proves that the below equation still converges within the neighborhood of the optimal concatenated, joint policies for all agents. The below ODE defines the derivative of the policy over time, and as the derivative approaches zero, the actor reaches a stable equilibrium (or a set of equilibrium points, since in a fully decentralized setting, the actor is a vector of joint policies for all agent, as described in Section 3) and thus, convergence of the actor is achieved.

$$\dot{\bar{\theta}} = \mathbb{E}_{\bar{s}_t \sim \bar{d}_\pi, \bar{a}_t \sim \bar{\pi}_\theta} \left[ \nabla \log(\bar{\pi}_\theta) * (\bar{Q}(\bar{s}_t, \bar{a}_t) - \bar{V}(\bar{s}_t)) \right] = 0 \tag{27}$$

### A.5 FULL PROOF OF THEOREM 1

Here, we derive the full proof of the Bayesian expansion of the policy (Theorem 1). As mentioned before, given the Bayesian nature of the $k$-level hierarchy, actions conceived at similar levels of $k$ can be reasonably assumed to be *independent*. Without loss of generality, we assume a scenario

with two cooperating agents $i$ and $j$ both with $k$ levels of rationalization. An important notational difference that will be used here is $p(.)$. Policies are conventionally considered state-conditional distributions of actions, where the action is the random variable. A *specific* action is usually either sampled from the distribution or the MAP action is selected. Here, we denote $p(X)$ to refer to the probability distribution of a particular random variable $X$. Note that, evaluating $p(X = x)$ returns the specific probability that $X = x$ (and not a distribution). In the following, we seek to determine the distribution of actions, for $i$, at each level $k$ using $p(.)$ notation. This is determined by marginalizing the level-$k$ action of agent $i$ across any *specific* action that $i$ or $j$ might take, which will be denoted as $x$ and $y$ respectively. Additionally, a specific observation in the observation space will be denoted as $z$. Accordingly, we can represent the probability of action $a$ for agent $i$ at time $t$ across levels of rationalization, $k \in \{0, 1, \cdots, K\}$ as shown in Equations 28-30.

$$p(a_t^{i,(0)}) = \sum_{z \in \mathcal{S}} p(a_t^{i,(0)} \mid o_t^i = z)p(o_t^i = z) \tag{28}$$

$$p(a_t^{i,(1)}) = \sum_{x \in \mathcal{A}} \sum_{y \in \mathcal{A}} p(a_t^{i,(1)} \mid a_t^{i,(0)} = x, a_t^{j,(0)} = y)p(a_t^{i,(0)} = x, a_t^{j,(0)} = y) \tag{29}$$

$$\vdots$$

$$p(a_t^{i,(k)}) = \sum_{x \in \mathcal{A}} \sum_{y \in \mathcal{A}} p(a_t^{i,(k)} \mid a_t^{i,(k-1)} = x, a_t^{j,(k-1)} = y)p(a_t^{i,(k-1)} = x, a_t^{j,(k-1)} = y) \tag{30}$$

At each level $k \geq 1$, the probabilities of actions are directly proportional to the probabilities of $i$'s and $j$'s actions at level $k-1$ actions. Next, we can re-arrange the following Bayes' rule, from Eq.31 to Eq. 32, to determine the probability of the joint distribution of the priors $p(a_t^{i,(k-1)}, a_t^{j,(k-1)})$:

$$p(a_t^{j,(k)} \mid a_t^{i,(k-1)}, a_t^{j,(k-1)}) = \frac{p(a_t^{i,(k-1)}, a_t^{j,(k-1)} \mid a_t^{j,(k)})p(a_t^{j,(k)})}{p(a_t^{i,(k-1)}, a_t^{j,(k-1)})} \tag{31}$$

$$p(a_t^{i,(k-1)}, a_t^{j,(k-1)}) = \frac{p(a_t^{i,(k-1)}, a_t^{j,(k-1)} \mid a_t^{j,(k)})p(a_t^{j,(k)})}{p(a_t^{j,(k)} \mid a_t^{i,(k-1)}, a_t^{j,(k-1)})} \tag{32}$$

Eq. 32 can be substituted back into the joint probability of the priors, $p(a_t^{i,(k-1)}, a_t^{j,(k-1)})$ in Eq. 30:

$$p(a_t^{i,(k)}) = \sum_{x \in \mathcal{A}} \sum_{y \in \mathcal{A}} p(a_t^{i,(k)} \mid a_t^{i,(k-1)} = x, a_t^{j,(k-1)} = y)p($$

$$a_t^{i,(k-1)} = x, a_t^{j,(k-1)} = y \mid a_t^{j,(k)})\frac{p(a_t^{j,(k)})}{p(a_t^{j,(k)} \mid a_t^{i,(k-1)} = x, a_t^{j,(k-1)} = y)} \tag{33}$$

Now, we move the term $p(a_t^{j,(k)})$ outside the summations, as it does not depend on the marginalization, and multiply both the numerator and denominator in Eq. 33 by the joint probability of the priors, $p(a_t^{i,(k-1)} = x, a_t^{j,(k-1)} = y)$, to arrive at Eq. 34:

$$p(a_t^{i,(k)}) = p(a_t^{j,(k)}) \sum_{x \in \mathcal{A}} \sum_{y \in \mathcal{A}} p(a_t^{i,(k)} \mid a_t^{i,(k-1)} = x, a_t^{j,(k-1)} = y)p($$

$$a_t^{i,(k-1)} = x, a_t^{j,(k-1)} = y \mid a_t^{j,(k)})\frac{p(a_t^{i,(k-1)} = x, a_t^{j,(k-1)} = y)}{p(a_t^{j,(k)} \mid a_t^{i,(k-1)} = x, a_t^{j,(k-1)} = y)p(a_t^{i,(k-1)} = x, a_t^{j,(k-1)} = y)} \tag{34}$$

Next, by simplification of the denominator we arrive at:

$$p(a_t^{i,(k)}) = p(a_t^{j,(k)}) \sum_{x \in \mathcal{A}} \sum_{y \in \mathcal{A}} p(a_t^{i,(k)} \mid a_t^{i,(k-1)} = x, a_t^{j,(k-1)} = y)p($$

$$a_t^{i,(k-1)} = x, a_t^{j,(k-1)} = y \mid a_t^{j,(k)})\frac{p(a_t^{i,(k-1)} = x, a_t^{j,(k-1)} = y)}{p(a_t^{j,(k)}, a_t^{i,(k-1)} = x, a_t^{j,(k-1)} = y)} \tag{35}$$

To simplify Eq.35, we first consider the transitivity of conditionals for the first two terms inside the summations. Note that, for a specific Bayes' tree in the form of our information graph shown in Fig. 1, we can write $\sum_{x \in \mathcal{A}} \sum_{y \in \mathcal{A}} p(a_t^{i,(k)} \mid a_t^{i,(k-1)}, a_t^{j,(k-1)}) p(a_t^{i,(k-1)}, a_t^{j,(k-1)} \mid a_t^{j,(k)}) = p(a_t^{i,(k)} \mid a_t^{j,(k)})$. Next, considering the summations, we marginalize both the numerator and the denominator in Eq. 35 across the joint action space. Accordingly, the numerator will simplify to 1 since we are summing the probabilities for all actions. Also, the marginalization across the joint action space to the denominator is simplified to $p(a_t^{j,(k)}, a_t^{i,(k-1)}, a_t^{j,(k-1)}) = p(a_t^{j,(k)})$. As such, by substituting these simplifications in Eq. 35, we arrive at Eq. 36.

$$p(a_t^{i,(k)}) = \frac{p(a_t^{j,(k)})}{p(a_t^{j,(k)})} p(a_t^{i,(k)} \mid a_t^{j,(k)}) = p(a_t^{i,(k)} \mid a_t^{j,(k)}) \tag{36}$$

In this form, we join parallels between $p(a_t^{i,(k)})$ and $\pi_{tot}^i(a_t^{i,(k)} \mid .)$. The $\pi_{tot}^i(a_t^{i,(k)} \mid .)$ is the probability distribution of the k-level action $a_t^{i,(k)}$, which according to the initial definition, is exactly the same as $p(a_t^{i,(k)})$ (same applies for $p(a_t^{j,(k)})$). Additionally, in the derivation of $p(a_t^{i,(k)} \mid a_t^{j,(k)})$, we used $p(a_t^{j,(k)} \mid a_t^{i,(k-1)}, a_t^{j,(k-1)})$; however, notice that this probability is exactly the same as $\pi_{com}^i(a_t^{j,(k)} \mid a_t^{i,(k-1)}, a_t^{j,(k-1)})$, therefore, we can make the following equivalencies and substitutions for both agents $i$ and $j$.

$$\pi_{tot}^i(a_t^{i,(k)} \mid .) = p(a_t^{i,(k)}) = \pi_{com}^i(a_t^{i,(k)} \mid a_t^{j,(k)}) \tag{37}$$

Now, considering that agent $i$ acted on its $k$-level reasoning and executed the MAP action, the log probability and its gradient of the action-distribution at the $k$-th level can be described as in Eq. 38-39:

$$\log(\pi_{tot}^i(a_t^{i,(k)} = \text{MAP} \mid .)) = \log(\pi_{com}^i(a_t^{i,(k)} = \text{MAP} \mid a_t^{j,(k)} = \text{MAP})) \tag{38}$$

$$\nabla \log(\pi_{tot}^i(a_t^{i,(k)} = \text{MAP} \mid .)) = \nabla \log(\pi_{com}^i(a_t^{i,(k)} = \text{MAP} \mid a_t^{j,(k)} = \text{MAP})) \tag{39}$$

As such, we arrive at the conclusion that was arrived upon near the end of Section 4.4: performing gradient ascent on $\pi_{tot}^i$ inherently increases $\pi_{com}^i(a_t^{i,(k)} \mid a_t^{j,(k)})$. This fact is used to support the mutual information bounds derived in Theorems 2 and 3, and is valid for any $k \geq 1$.

## A.6 Discussion on the Uniformity of Priors Assumption

Similar assumption of uniform priors as ours in Section 4.5 have been used previously by Prasad (2015) for the calculation of MI upper-bound. In our work, we defined $p(a_i)$ as a marginalization of the action-conditional policy, $\pi_{com}^i(a^i|a^j)$, across any potential $a^j$. The marginal's conceptual meaning here is similar to asking the question, "*What should the probability of $a^i$ be, if we did not know $a^j$?*" For a given action-conditional policy, we could expect $a^i$ to be uniformly random because the action-conditional policy is expected to only change the probability of a selected action based on the $k$-level reasoning with other agents. If there is no action to reason upon, the agent has no information with which to base its $k$-th action upon. It is important to note here that $p(a^i)$ is not a marginalization of both the state-conditional and action-conditional policies. Since $p(a^i)$ is only a marginalization of the action-conditional policy, we view the uniformly-random prior assumption as a reasonable design choice.

## A.7 Supplementary Results

In this section we provide our supplementary results. We start by analysing and interpreting agents' communicative policy in our fraudulent agent experiment in order to understand the effect of $k$-level rationalization for decision-making in this scenario. Next, we present the agent-level performances for InfoPG and compare with the baselines in all three evaluation environments. Next, we present an scalability analysis in the Pistonball environment to investigate robustness to larger number of agents. Eventually, we conclude this section by presenting a list of takeaways.

### A.7.1 Policy Interpretation for the Fraudulent Agent Experiment

In this section, we present an analysis and interpretation of agents' communicative policy in our fraudulent agent experiment in order to understand the effect of $k$-level rationalization for decision-making in this scenario. We test the learnt policy at convergence using Adv. InfoPG in the fraudulent

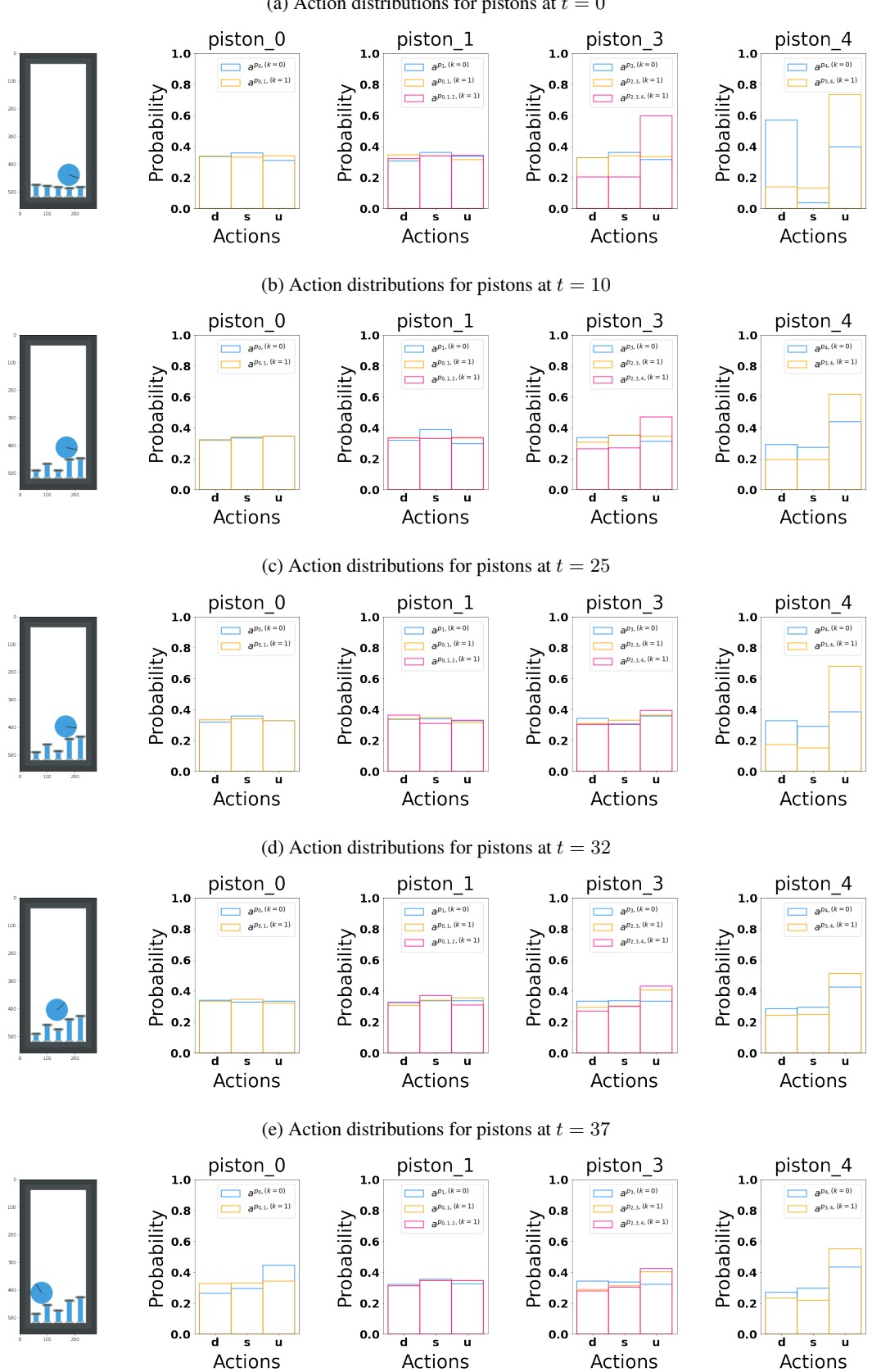

Figure 4: Action distributions of piston agents across 37 timesteps for the fraudulent agent experiment introduced in Section 6. Note that Piston 2 (not displayed) is the fraudulent agent with untrainable random policy.

agent experiment, presented in Section 6. The results are shown in Figure 4 in which, we present the action distributions of agents before and after rationalizing their decisions with their neighbors through $k$-level reasoning. For each action distribution graph, the Y-axis is the probability of an action and the X-axis represents actions, where *u=move up*, *d=move down*, and *s=stay* constant.

In Figure 4 we present sample illustrations of a test run in a BGP scenario for $t = 0, 10, 25, 32, 37$, from start to the end of the episode. At $t = 0$ (Fig. 4a), we can see that the episode starts with the ball on top of the right-most pistons (i.e. pistons 3 and 4). Note that pistons are indexed left to right. With $k = 1$ reasoning, we show piston 3's rationalization for its action decision in three phases: first, $a^{p3,(k=0)}$, is the naive $k = 0$ action (blue); second, $a^{p2,3,(k=1)}$, is the $k = 1$ rationalization that incorporates piston 2's random action (orange), and, third, $a^{p2,3,4,(k=1)}$, is the $k = 1$ rationalization that incorporates both piston 2's random action and piston 4's $k = 0$ action into piston 3's $k = 1$ action decision (pink). Using a similar notation convention, $a^{p3,4,(k=1)}$ is piston 4's action rationalization given piston 3's $k = 0$ action rationalization. Now, notice that the spread of $a^{p3,(k=0)}$ at $t = 0$ is relatively uniform (blue), and given that piston 2 is randomly moving, $a^{p2,3,(k=1)}$ remains unchanged (orange). This indicates piston 3 has learned to *ignore* piston 2, which is the fraudulent agent. Additionally, notice that $a^{p4,(k=0)}$ at $t = 0$ is bimodal and has relatively equal probabilities of moving either *up* or *down*; however, we can see for both pistons 3 and 4, after rationalizing their actions with each other at $k = 1$, both action distributions become unimodal and tend towards moving up, which is the desired action for moving the ball to left.

This coordination demonstrates an interesting strategy; piston 3 and piston 4 have learned that coordinating actions with the randomly moving piston 2 is not desirable and therefore, they seek to move the ball as high as possible, and toss it over piston 2. Empirical proof of this behavior can be seen by the continuation of the spread of distributions at $t = 10$. At $t = 25$ a distinct change occurs; piston 3's action distribution, after $k = 1$ rationalization with piston 4, becomes uniform again, while Piston 4 is still unimodal and tending up. We believe this behavior shows that piston 3 and 4 have realized the strategy to bypass piston 2 is to "launch" the ball over piston 2, which can be accomplished by piston 4 moving up, while piston 3 remains stable, effectively creating a leftward force for the ball to move left. At $t = 32$ we can see the "launching" is performed, and here the action distributions of both piston 3 and 4 become relatively uniform again (Note that actions of piston 3 and 4 do not matter at this point since they are not directly located under the ball and therefore, do not receive a reward for their actions). From $t = 32$ to $t = 37$, the ball traverses over pistons 0 and 1; however, note that piston 0 and 1 will not need to move the ball too much, since the ball already has leftward momentum. Accordingly, piston 0 and 1 only need to coordinate to create a leftward ramp to facilitate ball's movement. As such, both piston 0 and piston 1 follow relatively uniform distributions after k=0 and 1 rationalization. At $t = 37$, the ball is over piston 0 and has reached the left wall, which denotes winning and end of the episode.

In summary, we show two key behaviors learnt by agents through our Adv. InfoPG in the fraudulent agent experiment: 1) Piston 2 is untrustworthy and thus, coordinating with this agent is not desired, which leads to unchanged action-distributions for Pistons 3 and 1 after iterated $k$-level reasoning with this fraudulent agent. 2) Pistons 3 and 4 learn to avoid the fraudulent agent (piston 2) by "launching" the ball over it, giving the ball a leftward momentum to reach the left wall.

### A.7.2 A QUALITATIVE ANALYSIS FOR BOUNDED RATIONALITY

The postulate of $k$-level reasoning is that higher levels of $k$ should allow for deeper decision rationalization and therefore better strategies. In this section, we qualitatively investigate *different examples of intelligent behavior induced by varying bounds of rationality*. To address this, we specifically further investigate InfoPG's results in the MultiWalker and StarCraft II (SC2) domains due to their complex mechanics and multi-faceted objectives. In the following, we first present our qualitative analysis for SC2 and Multiwalker, respectively.

**SC2 –** Our qualitative analysis in SC2 is a demonstration of how bounded rationality and higher levels of iterated reasoning benefits performance. In SC2, agents are positively rewarded for shooting and killing enemy agents, and are negatively penalized for getting shot at. Therefore, a locally optimal strategy is to run away from the enemy team to avoid any negative penalties, while a globally optimal strategy is to kill and eliminate all the enemy agents to achieve high positive rewards.

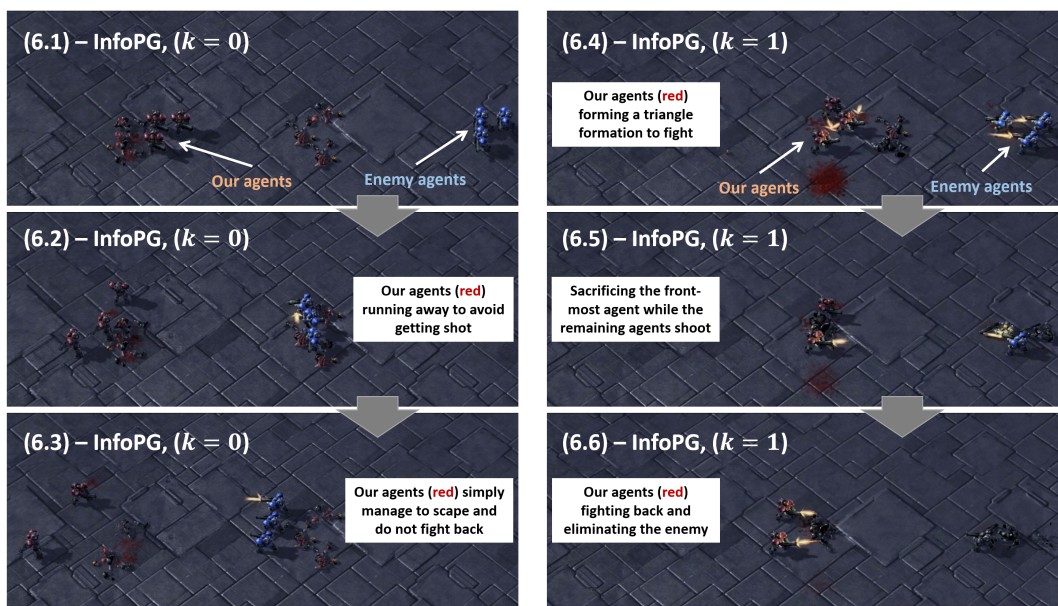

Figure 5: Comparing the learned policies by InfoPG at convergence in the SC2 domain with $k = 0$ and $k = 1$. With $k = 0$ (Fig. 5.1–5.3), the naive agents disregard other agents actions and simply learn to run away to avoid the negative penalties of getting shot at. This is while the more sophisticated agents with $k = 1$ (Fig. 5.4–5.6), learn more strategic policies to work together to eliminate the enemy team and achieve high positive rewards.

Fig. 5 shows our qualitative results for analyzing the effects of assuming bounded-rational agents and iterated reasoning in the SC2 domain. We compare the learned policies by InfoPG at convergence in the SC2 domain with $k = 0$ and $k = 1$.

At $k = 0$ of the rationalization hierarchy, the fully naive and non-strategic level-0 agents choose actions while completely disregarding other agents actions (i.e., have zero-order beliefs). As such, for a level-0 policy, we expect to observe that agents simply run away from the enemy to avoid getting shot at, since a single agent does not believe (zero-order belief) it can overcome the enemy team. As seen in Fig. 5.1–5.3, the naive agents expectedly only learn to run away to avoid the negative penalties of getting shot at. This fleeing behavior allows agents to maintain a reward of zero, as shown in Fig. 2, indicating successful escape and convergence to the locally optimum solution.

At level $k = 1$, each agent is now more sophisticated and believes that the other agents have a level-0 policy and takes actions according to that. In this case, we observe a vastly different behavior. As shown in Fig. 5.4–5.6, agents here learn more strategic policies to work together to eliminate the enemy team and achieve high positive rewards. The agents begin a triangle-like formation towards the enemy (Fig. 5.4). Enemy agents then begin to shoot the closest opposing player at the front of the triangle formation. The other two agents in the team use this opportunity and start firing at the enemy team while they shoot the front-most agent. As such, the remaining two agents manage to kill and eliminate the enemy team. Through reasoning their level-1 actions based on their teammates level-0 action, InfoPG agents learn a sacrificial technique of exposing one agent as bait, which allows the agents to converge to the globally optimum solution of killing the entire enemy team. This is also reflected in Figure 2, where the $k = 1$ InfoPG achieves the highest cumulative rewards.

**Multiwalker –** Our qualitative analysis in Multiwalker is another demonstration of how higher levels of iterated reasoning benefits performance. There are two objectives that the walkers need to satisfy: (1) stabilization (both the package and the walkers) and (2) moving forward. Stabilization of the package and the walkers are the primary goals, since dropping the package, or falling, results in failing the game with a penalty. Walking (i.e., moving forward to the right) is an additional goal, since every forward step yields a small proportional reward. Ultimately, the walkers should aim to achieve both stabilization and walking at the same time, which is the globally optimum solution.

Fig. 6 shows our qualitative results for analyzing the effects of assuming bounded-rational agents and iterated reasoning in the Multiwalker domain. At level $k = 1$, each walker believes that the other walker has only a level-0, non-rational policy. We observe in Fig. 6-(a) that, with InfoPG and by

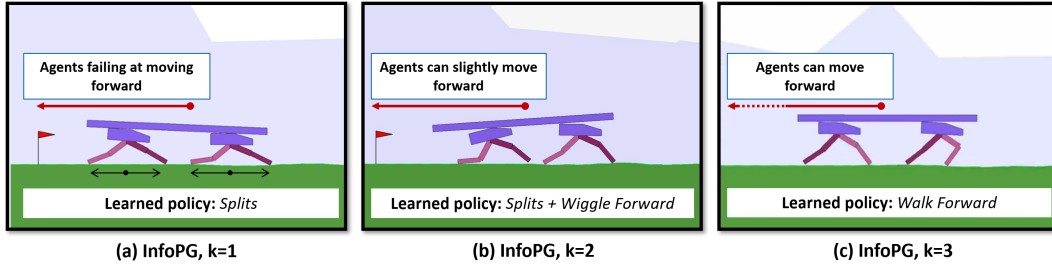

| (a) InfoPG, k=1 | (b) InfoPG, k=2 | (c) InfoPG, k=3 |

Figure 6: Comparing the learned policies by InfoPG in the Multiwalker with $k = 1$, $k = 2$, and $k = 3$, Fig. 5a-5c, respectively. With $k = 1$, agents only learn to perform a split to balance the package on top and avoid falling. This is while with $k = 3$ agents learn to quickly walk forward. The middle stage of rationalization, $k = 2$, achieves an in-between policy where agents split to balance, but also wiggle forward slowly.

$k = 1$, the walkers solve only the stabilization problem by learning to do the "splits", which is a locally optimum solution. The walkers create a wide base with their legs and simply hold the package statically with no forward progress (evidenced by the starting red flag). This technique requires some degree of coordination since the walkers have to do the splits synchronously at the beginning of the experiment; however, this is not nearly a complex enough strategy to achieve any positive rewards. Looking at the reward convergence in Fig. 2, $k = 1$ achieves converges to the locally optimum solution and achieves a reward of 0, since the walkers do not get any positive reward for moving nor do they get any negative reward for dropping the package or falling.

At level $k = 2$, each walker now believes that the other walker has a level-1, bounded-rational policy. Intuitively, assuming a more sophisticated policy for a teammate should lead to a better overall strategy, since the best-response solution to such sophisticated policy needs a certain level of sophistication Gershman et al. (2015); Ho & Su (2013). We observe in Fig. 6-(b) that, as expected, the learned policies at level $k = 2$ of rationalization still includes performing the "splits" for balancing while agents also learned to wiggle forward slowly and receive some positive reward.

As we increase the rationalization depth from $k = 3$, we see in Fig. 6-(c) that the walkers not only stabilize the package, but also start moving forward (evidenced by the starting red flag out of frame) with much more sophisticated strategies. The left-most walker learns to generate forward momentum and walk more quickly than the front walker, which learns to walk more slowly and maintain the stability of the package. Note that this illustrates the idea of role allocation, which is a relatively complex strategy and indicative of higher levels of intelligence achieved through assuming sophisticated, level-2 teammates. The walkers learn to coordinate their movements, because if the left-most walker makes too jerky of a forward movement, the right-most walker adjusts by staying more static to stabilize the package. In Fig. 2, the collective strategy at $k = 3$ can achieve rewards as high as +10, which is the globally optimum solution.

### A.7.3 SCALABILITY ANALYSIS: PISTONBALL

Here, we investigate InfoPG's robustness to larger number of interacting agents in the Pistonball environment. For this experiment, We selected our best-performing model, Adv. InfoPG, and the best-performing baseline, MOA (Jaques et al., 2019), from our primary results in Section 6. We increased the number of agents from five to ten and kept the communication range to be the same (i.e., one piston on each side). The results are presented in Fig. 7. As shown, Adv. InfoPG outperforms MOA in both maximizing average individual and team reward performances during training.

InfoPG considers two way communication with each of its neighbors (there are $|\Delta|$ neighbors which are communicated with $k$ times). If $\mathcal{D}$ is the dimension of the communicated $k$-level action-vector, the bandwidth of input and output communication channels is $\Theta(2|\Delta|k\mathcal{D})$, where each communication channel is $\Theta(|\Delta|k\mathcal{D})$. We leave the choice of $|\Delta|$, $k$, and $\mathcal{D}$ to be hyper-parameters, all of which can be lessened as the number of agents increase to inhibit computational complexity issues.

### A.7.4 AGENT-WISE PERFORMANCE COMPARISON

As mentioned, the objective in a *fully* decentralized domain is to maximize the average return by each individual agent, such that the obtained cumulative team reward is also maximized. We have show in

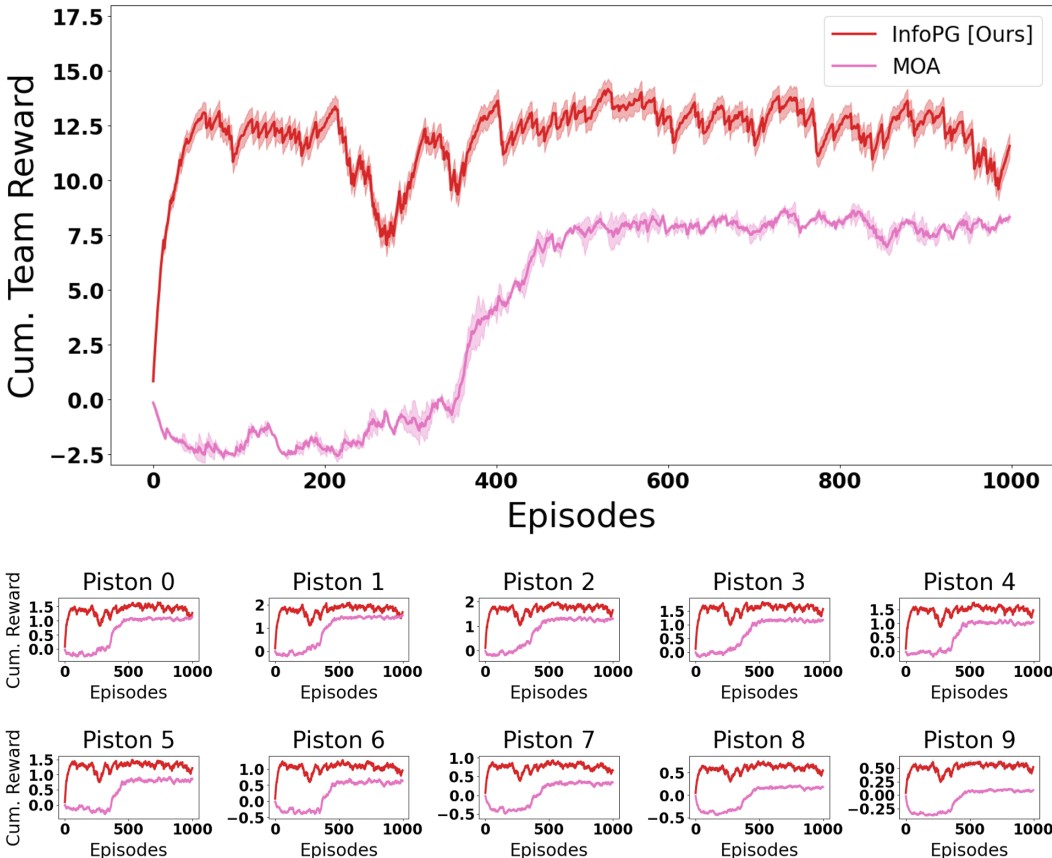

Figure 7: Scalability comparison between Adv. InfoPG and the best-performing baseline, MOA (Jaques et al., 2019), in the Pistonball domain with ten interacting agents. Adv. InfoPG outperforms MOA in both maximizing average individual and team reward performances.

our primary results in Section 6 that InfoPG outperforms all baselines across all three domains in achieving higher cumulative team reward. Here, we present the agent-level reward performances for InfoPG and compare with the baselines across three domains. The results are presented in Fig. 8, where sub-figures 8a-8d represent the individual agent performances in Co-op Pong, Pistonball, Multiwalker and StarCraft II (3M), respectively. As shown, our InfoPG and its MI-regularizing variant, Adv. InfoPG, continually outperform the other baselines in maximizing achieved individual rewards for agents. Specifically, in all graphs for Adv. InfoPG, all agents maximize individual rewards over time, and Adv. InfoPG achieves SOTA across all baselines.

## A.8  EVALUATION ENVIRONMENTS: ADDITIONAL DETAILS AND PARAMETERS

Here we provide additional details regarding the employed evaluation environments for training and testing InfoPG and cover the associated environment parameters related to our experiments. An instance of the four environments are presented in Fig. 9.

**1) Cooperative Pong (Co-op Pong) (Terry et al., 2020)** – The objective in this game is to keep a pong ball in play for as long as possible between two co-operating paddles. To fit the MAF-Dec-POMDP paradigm, agents must receive individualistic rewards. In the Co-op Pong domain, each paddle receives a reward of $+1$ if it hits an incoming pong ball successfully and a penalty of $-1$ if it misses. The game ends either when a paddle misses or if $max\_cycles = 300$ cycles have elapsed. Therefore, in order to continue receiving positive rewards of $+1$, paddles are implicitly encouraged to cooperate to maximize their accumulated rewards. For an episode of the game, the pong ball was set to move at a velocity of $15[\frac{pixels}{sec.}]$ while the paddles move slightly slower at $13[\frac{pixels}{sec.}]$. Since the ball moves faster than the paddles, the paddles require "forecasting" their intended position when the pong ball comes into their field-of-view (FOV), which is a $280 \times 240$ RGB image. An important facet of this environment is the time-delayed nature of the actions. Consider a scenario when the left-side

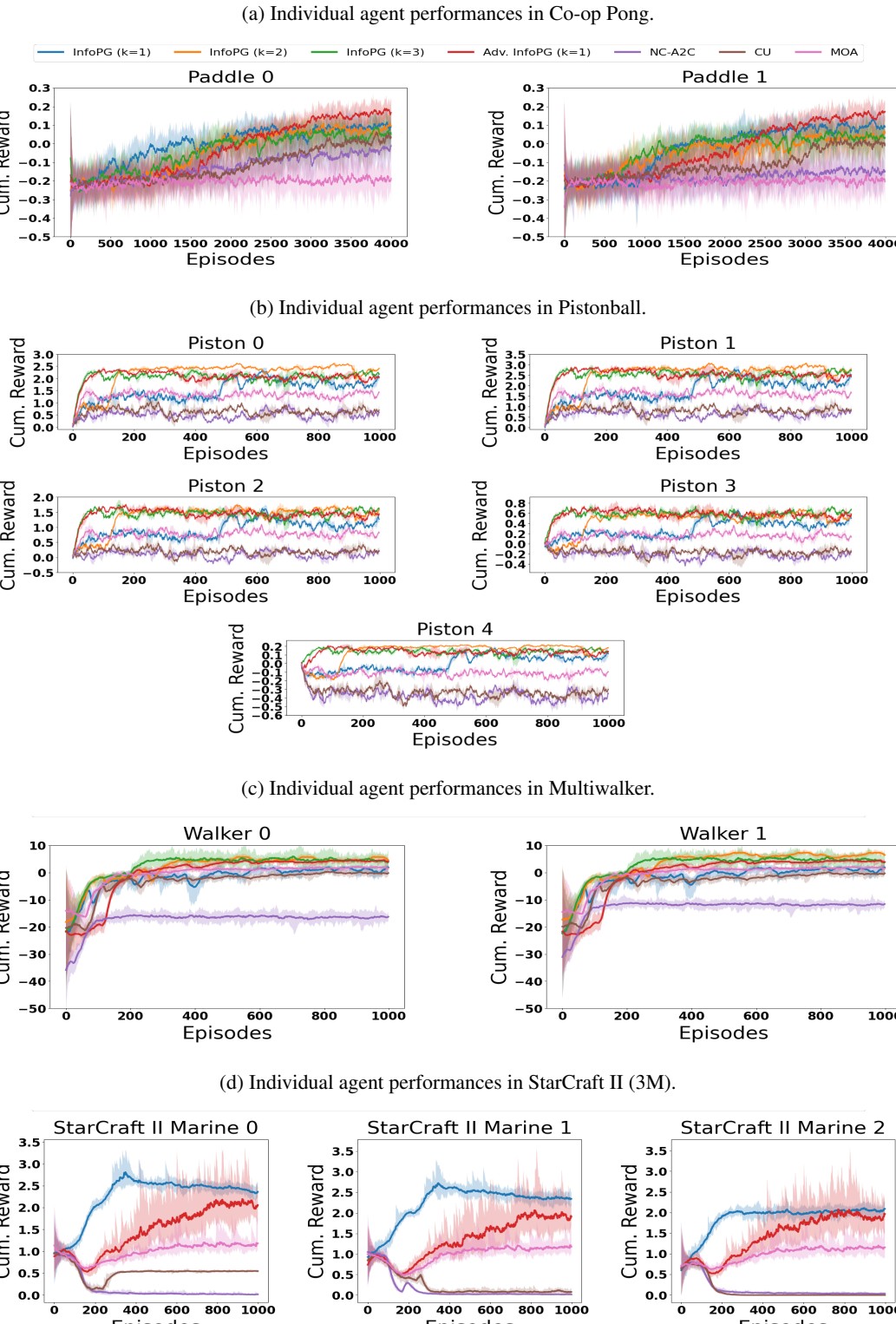

Figure 8: Individual rewards obtained by each individual agents across episodes as training proceeds in the three evaluation environments. Our Adv. InfoPG continually outperforms all baselines (Wen et al., 2019; Jaques et al., 2019; Zhang et al., 2018; Sutton & Barto, 2018) across all domains.

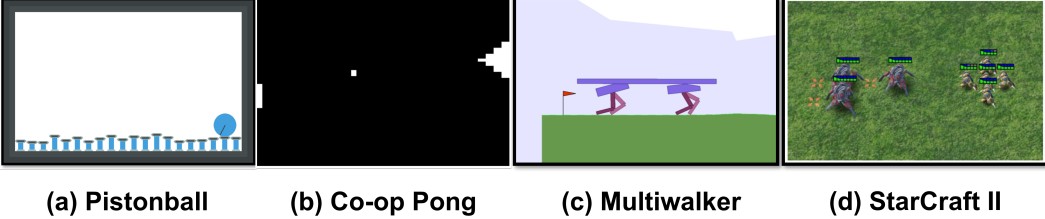

| (a) Pistonball | (b) Co-op Pong | (c) Multiwalker | (d) StarCraft II |

Figure 9: Instances of the utilized multi-agent cooperative environments. Domains are parts of the PettingZoo (Terry et al., 2020) MARL research library and can be accessed online at `https://www.pettingzoo.ml/envs`. The StarCraft II (Vinyals et al., 2017), can be accessed from Deepmind's repository available online at `https://github.com/deepmind/pysc2`.

paddle hits the ball at $t_1$; this means that the right-side paddle will receive the ball at minimum, at $t_2 = t_1 + \frac{280}{15}$. This measure is an underestimation, since the pong ball will not likely move in a straight line drive (i.e. it may hit the sides of the walls) but it illustrates the point that the action of the left-side paddle at $t_1$ is particularly relevant to the action of the right-side paddle at $t_2$; however, the action of the left-side paddle at $t_2$ is not particularly relevant to the right-side.

Therefore, in our experiments, to account for the time-delay in the action information, a "hub" was designed where the action at the time of the last "hit" is shared to the opposing paddle, and a zero-vector otherwise. This procedure was applied for both InfoPG and MOA Jaques et al. (2019) as to fairly evaluate the communicative algorithms. In the case of MOA, the time-delayed action was sought to be "predicted" by the opposing paddle's model of agent.

**2) Pistonball (Terry et al., 2020)** - The goal in the this environment is to move a ball from right side of the screen (e.g., right wall) to the left side of the screen (e.g., left wall) by activating/deactivating a team of five vertically moving piston agents. The ball has momentum in motion and is elastic. In order to encourage robustness, the ball was randomly placed on the pistons with a friction factor of $0.3$, mass of $0.75$, and a relatively high elasticity factor of $1.5$. An episode of the game ends if agents move the ball to the left wall or after $max\_cycles = 200$ cycles have elapsed. Each agent's observation is an RGB image of size $457 \times 120$ covering the two pistons (or the wall) around an agent and the space above them. Each piston receives an individual reward which is a combination of how much the corresponding agent *directly* contributed to moving the ball to left (i.e., with a value of $\mathcal{X}_t^{ball} - \mathcal{X}_{t-1}^{ball}$, where $\mathcal{X}_t$ represents the center position of the ball along the X-axis at time $t$), and a negative time penalty of $-0.007$ per timestep. A piston is considered to be contributing directly to moving the ball to left, if it is directly below any part of the ball. Given the ball's radius of 20 pixels, at each timestep, three pistons can be directly under the ball. Agents win an episode of the game if they can coordinate efficiently to move the ball to the left wall within allowed maximum steps.

**3) Multiwalker (Gupta et al., 2017; Terry et al., 2020)** - The objective in this continuous-space environment is for a team of two bipedal robots to carry a heavy package cooperatively and walk as far right as possible. The weight of the package depends on its length which is determined by the number of agents. We note that, the two-agent case is the most challenging scenario in Multiwalker. Each robot exerts a variable force on two joints in its two legs, and therefore, the action-space is of dimension four with values in range $(-1, 1)$. The bipedal robots receive local rewards related to individual balance and stability of their hull. The reward function includes a reward of $+1$ for a scaled forward displacement of each bipedal robot's hull. We set the maximum number of allowed steps to $max\_cycles = 500$ cycles, which would terminate at any point if a bipedal or the package falls. If a bipedal robot falls, it will individually receive a penalty of $-10$, and, if the package falls on the ground, each bipedal receives a penalty of $-100$. Each agent receives a 31-dimensional observation vector. The first 24 elements of the observation vector represent the bipedal robot's internal kinematics and the rest (i.e, elements 24-31) relate to LIDAR observations of the package position as well as the position of the adjacent bipedals.

**4) StarCraft II** (Vinyals et al., 2017) (*The 3M (three marines vs. three marines) challenge*) - The goal in this domain is for a team of three friendly marines to find, shoot, and kill three enemy marines as soon as possible, without dying or getting hit (Seraj et al., 2020; 2019). This domains is more challenging than the previous ones since the state-space is larger and the communication graph is time-varying. In this challenge, marines can move in four primitive directions and shoot an enemy

marine within distance (i.e., multiple enemy marines can be in vision at once), and therefore, the action-space dimension is also larger than the previous domains. Agents get negatively rewarded when they are shot by enemy marines and are positively rewarded when shooting enemy marines. 3M presents a fundamentally more challenging environment than Pistonball, Multiwalker, and Pong, as the state space is much larger, and agents are allowed to move in 2D over a large gameplay arena. The action space is also larger (size 8) as agents can choose between: no action, moving in 4 directions, and shooting any one out of 3 enemy marines. Additionally, agents have a time-varying communication graph (different from the other domains), because friendly marines move in and out of the line of sight.

### A.9 TRAINING AND EXECUTION DETAILS

Under the MAF-Dec-POMDP paradigm, each agent $i \in \mathcal{N}$ is equipped with its own optimizer and policy $\pi^i_{tot}$ which consists of an encoding policy $\pi^i_{enc}$ and a communicative policy $\pi^i_{com}$ (each parameterized by $\theta^i$). The encoding policy can be represented using a feed-forward neural network, and the communicative policy can be represented by any class of Recurrent Neural Networks (RNNs), such as the Gated Recurrent Unit (GRU) or Long Short-Term Memory (LSTM). For computational efficiency, we chose to use a GRU or simplistic RNN architecture.

Additionally, while we formulaically denote actions for level $k$ as $a^{i,(k-1)}_t$, in execution we represent these actions as finite-dimensional vectors to maximize information during inference. The size of these vectors are known as *policy latent size* in the provided hyperparameter tables. This parameter (also shared by other baselines) refers to the size of the latent vector prior to the final Softmax output layer. During the encoding stage of InfoPG, each agent, $i$, receives an observed state vector, $o^t_i$, and encodes an action vector $a^{i,(0)}_t$, using $\pi^i_{enc}$. During $k$-level communication, each agent receives the action vectors of neighboring agent $j \in \Delta^i_t$ from level $k-1$ and performs a forward-pass on the RNN, where the initial cell state is $a^{i,(k-1)}_t$. The action probabilities for the discrete domains (i.e., Co-op Pong and Pistonball) are outputted by the feed-forward network where the last layer size is equal to the size of the action space and a Softmax activation. Note that for our continuous action-space domain, Multiwalker, the final Softmax activation function is replaced with the Tanh activation. Neighboring agents are determined using the adjacency graph $\mathcal{G}_t$, and a distance hyper-parameter specifying how "far" agent $i$ can communicate (i.e., communication range). $\mathcal{G}_t$ is an undirected time-varying graph, and as agents perform actions and change their relative position (depending on the domain), the edges $\mathcal{E}_t \subseteq \{(i,j) : i, j \in \mathcal{N}, i \neq j\}$ are updated for the next timestep. This process is carried out until convergence of the cumulative rewards of all $N$ agents.

**Specifics for Co-op Pong –** The input observation in this domain, a $280 \times 240$ RGB image, contains information about where the pong ball exists in the FOV of each paddle. Since the ball is in motion, we found higher performance could be achieved by setting the observation at time, $t$, to be the difference between the observation at $t$ and $t-1$. As such, we encoded the input observation to represent information about not just the position, but also the velocity of the ball. This procedure was maintained for all baselines.

Another property we found critical to the performance of InfoPG agents in Co-op Pong was the *type* of RNN for $\pi^i_{com}$. In Pong, rewards are rather sparse, since paddles only receive feedback when they hit or miss a ball, while in other times and when ball is traversing the screen (which is the majority of the time spent in the game) no feedback is received from the environment. Accordingly, we leveraged curriculum learning (Bengio et al., 2009) such that we let agents first learn the mechanics of hitting the pong ball and then, learn the benefit of communication. We achieved this behavior by using a simple RNN (we distinguish this with VRNN for Vanilla RNN) cell for $\pi^i_{com}$, where the initial weight matrix $W_{ih}$ was set to the identity matrix and all other parameters were set to a small constant. This way, we effectively make the output of the $\pi^i_{com} = \pi^i_{enc}$ at the beginning episodes of training, while as time elapses, the weight matrix is optimized to incorporate actions from the neighboring paddle.

**Specifics for Pistonball –** The agents each receive a $457 \times 120$ RGB image as their observation input. In order to minimize feature size, each observation was first cropped to a size of $224 \times 224$, normalized and inputted into a pre-trained AlexNet model. AlexNet (Krizhevsky et al., 2012) is a CNN that takes in images and outputs probability scores of classes. In our experiments, we utilized the first four intermediate layers of AlexNet to produce rich feature observations to the input of the encoding policy. This procedure was applied to all baselines.

**Hardware Specifics –** All experiments were conducted on an NVIDIA Quadro RTX 8000 with approximately 50 GB of Video Memory Capacity.

**Training Hyperparameters –** We present the training hyperparameters in our implementations and experiments across methods and all three environments in Tables 2-6.

Table 2: Co-op Pong Training Hyperparameters.

| Experiments | Pong | | | | | |
|---|---|---|---|---|---|---|
| | InfoPG | Adv InfoPG | NC-A2C | CU | MOA | PR2-AC |
| Learning Rate | 4e-4 | 4e-4 | 4e-4 | 4e-4 | 4e-4 | 4e-4 |
| Size of Latent Vector | 30 | 30 | 30 | 30 | 30 | 30 |
| Type of Com. Network | VRNN | VRNN | - | - | GRU | - |
| Epochs | 4000 | 4000 | 4000 | 4000 | 4000 | 4000 |
| MOA Loss Weight | - | - | - | - | 0.1 | - |
| Discount Factor | 0.95 | 0.95 | 0.95 | 0.95 | 0.95 | 0.99 |
| Batch Size | 16 | 16 | 16 | 16 | 16 | 16 |
| Max Gradient Norm | 10 | 10 | 10 | 10 | 10 | 10 |
| Replay Buffer Size | - | - | - | - | - | 1e5 |
| Number of Particles | - | - | - | - | - | 16 |

Table 3: Pistonball Training Hyperparameters.

| Experiments | Pistonball | | | | | |
|---|---|---|---|---|---|---|
| | InfoPG | Adv. InfoPG | NC-A2C | CU | MOA | PR2-AC |
| Learning Rate | 1e-3 | 1e-3 | 1e-3 | 1e-3 | 1e-3 | 1e-3 |
| Size of Latent Vector | 20 | 20 | 20 | 20 | 20 | 20 |
| Type of Com. Network | GRU | GRU | - | - | GRU | - |
| Epochs | 1000 | 1000 | 1000 | 1000 | 1000 | 1000 |
| MOA Loss Weight | - | - | - | - | 1.0 | - |
| Discount Factor | 0.99 | 0.99 | 0.99 | 0.99 | 0.99 | 0.99 |
| Batch Size | 4 | 4 | 4 | 4 | 4 | 4 |
| Max Gradient Norm | 0.75 | 0.75 | 0.75 | 0.75 | 0.75 | 4.0 |
| Replay Buffer Size | - | - | - | - | - | 1e5 |
| Number of Particles | - | - | - | - | - | 16 |

Table 4: Multiwalker Training Hyperparameters.

| Experiments | Multiwalker | | | | | |
|---|---|---|---|---|---|---|
| | InfoPG | Adv. InfoPG | NC-A2C | CU | MOA | PR2-AC |
| Learning Rate | 4e-4 | 4e-4 | 4e-4 | 4e-4 | 4e-4 | 4e-4 |
| Size of Latent Vector | 30 | 30 | 30 | 30 | 30 | 30 |
| Type of Com. Network | GRU | GRU | - | - | GRU | - |
| Epochs | 1000 | 1000 | 1000 | 1000 | 1000 | 1000 |
| MOA Loss Weight | - | - | - | - | 0.1 | - |
| Discount Factor | 0.95 | 0.95 | 0.95 | 0.95 | 0.95 | 0.95 |
| Batch Size | 16 | 16 | 16 | 16 | 16 | 16 |
| Max Gradient Norm | 5 | 5 | 5 | 5 | 5 | 5 |
| Replay Buffer Size | - | - | - | - | - | 1e5 |
| Number of Particles | - | - | - | - | - | 16 |

Table 5: StarCraft II Mini-game (The 3M Challenge) Training Hyperparameters.

| Experiments | StarCraft II (3M Challenge) | | | | | |
|---|---|---|---|---|---|---|
| | InfoPG | Adv. InfoPG | NC-A2C | CU | MOA | PR2-AC |
| Learning Rate | 1e-4 | 1e-4 | 1e-4 | 1e-4 | 1e-4 | 1e-4 |
| Size of Latent Vector | 50 | 50 | 50 | 50 | 50 | 50 |
| Type of Com. Network | GRU | GRU | - | - | GRU | - |
| Epochs | 1000 | 1000 | 1000 | 1000 | 1000 | 1000 |
| MOA Loss Weight | - | - | - | - | 0.1 | - |
| Discount Factor | 0.99 | 0.99 | 0.99 | 0.99 | 0.99 | 0.99 |
| Batch Size | 64 | 64 | 64 | 64 | 64 | 64 |
| Max Gradient Norm | 6 | 6 | 6 | 6 | 6 | 6 |
| Replay Buffer Size | - | - | - | - | - | 1e5 |
| Number of Particles | - | - | - | - | - | 16 |

Table 6: Pistonball Training Hyperparameters for the fraudulent agent experiment.

| Experiments | Fraud Pistonball | | |
|---|---|---|---|
| | InfoPG | Adv InfoPG | MOA |
| Learning Rate | 1e-3 | 1e-3 | 1e-3 |
| Size of Latent Vector | 20 | 20 | 20 |
| Type of Com. Network | GRU | GRU | GRU |
| Epochs | 1000 | 1000 | 1000 |
| MOA Loss Weight | - | - | 1.0 |
| Discount Factor | 0.99 | 0.99 | 0.99 |
| Batch Size | 2 | 2 | 2 |
| Max Gradient Norm | 0.5 | 0.5 | 0.5 |

