# OpenReview forum: "Iterated Reasoning with Mutual Information in Cooperative and Byzantine Decentralized Teaming"
_ICLR.cc/2022/Conference — ICLR 2022 Poster_

### Official Review · Reviewer_7nod · 2021-11-01

**Correctness:** 4
**Technical Novelty And Significance:** 4
**Empirical Novelty And Significance:** 4
**Recommendation:** 8
**Confidence:** 3

**Main Review:**

## Strong points

I find it very innovative
1. to reformulate the policy $p(.|s)$ to be $p(.|a^{j,k}, a^{i,{k-1}},  a^{i,{k-1}}, ...)$
2. to use the K-level Bayesian tree to represent the interaction between agents.

The experiments are abundant

## Issues
1. Line 12 of Algorithm 1, how the MAP sample is carried out? For discrete actions, one can choose the max posterior action by enumeration, what about continuous actions?

2. Line 19 of Algorithm 1: which equation does this correspond to?
In the explanation text, line 19 is not mentioned.

3. What is $J(\theta)$ exactly in Eqn (2) and (15)? Could you elaborate on that?

4.  In Appendix A.5 Full proof of Theorem 1, page 17, paragraph between equation (35) and eqn. (36),  for a Bayesian tree, why
  $\sum_x \sum_y p(x'|x,y)p(x,y|y')=p(x'|y')$?


**Summary Of The Paper:**

The paper proposed  Info-PG, by reformulating the individual policy conditioned on state or partial observation to conditional on actions of other agents and with K-level rationalizability from cognitive hierarchy theory, the policy is conditioned recursively on lower levels of actions from all agents.
The optimization of the policy is in the same form of policy gradient algorithm.
The author showed that with reward shaping or applying relu on advantage, optimizing the reformulated policy is implicitly maximizing lower bound of mutual information across agents' policies.
In the case of Byzantine General's Problem where at least one agent is not cooperative, which could be detrimental to multi-agent learning, the author showed that by removing the relu on advantage, the policy gradient on the reformulated policy lead to tuning the upper bound of mutual information. The mutual information between the agent in question and the fraudulent agent will has its upper bound tuned down in case of negative reward, thus the fraudulent agent won't reduce the uncerntainty of policy of the agent in question, thus won't poison the multi-agent learning.
The algorithm is evaluated on a variety of multi-agent decentralized learning environments with promising results.



**Summary Of The Review:**

In general the paper is innovative and solid, there are several issues as in the main review section that need to be addressed.

---

> ### Author Response · Authors · 2021-11-14
> **Official Response to Reviewer 7nod**
>
> We appreciate the reviewers time and constructive questions. Below we respond to all the questions one-by-one.
>
> ### Strong points
> Thank you for pointing out the strength in our work.
>
> ### Issues
> - Thank you for pointing this out. Theoretically, the MAP action for a continuous action is just the mean action without any standard deviation from the normal distribution. We added this description to the Appendix on page 15, section A.2 for our readers.
>
> - Line 19 is showing the gradient of our value estimate. The loss for our value estimate V(s_t) is the cumulative discounted rewards subtracted by the true value of s_t, which is the advantage.  Therefore, the gradient is the sum of squared individual advantages. We added this description for line 19 of the algorithm to the text on page 14, second paragraph under the algorithm.
>
> - $J(\theta)$ describes the individualistic objective function for an agent $i$, parametrized by $\theta$, which seeks to learn a policy that maximizes the cumulative future rewards. The term in Eqn $(2)$ and $(15)$ represents the gradient of the objective function with respect to its parameter, calculated by the policy gradient theorem (Sutton & Barto, 2018).
>
> - We exploit the properties of our specific Bayes net, shown in figure 1, for this property to compute the conditional probability. $p(x'\vert y')$ is achieved by marginalization over $x$ and $y$ in the conditional and joint conditional probabilities. A straightforward demonstration of this property is presented here: https://ocw.mit.edu/courses/electrical-engineering-and-computer-science/6-034-artificial-intelligence-fall-2010/tutorials/MIT6_034F10_tutor06.pdf
>
> ### Summary of the Review
> We appreciate the reviewer’s constructive questions that helped us improve our manuscript. We  provided answers to the questions and made appropriate edits to the paper where needed.

---

### Official Review · Reviewer_SyxF · 2021-11-02

**Correctness:** 3
**Technical Novelty And Significance:** 2
**Empirical Novelty And Significance:** 2
**Recommendation:** 6
**Confidence:** 3

**Main Review:**

# Strengths
- The paper is well written. For the most part, each section makes its points clearly and the paper is easy to follow.
- Insofar as it is defensible (see Weaknesses below), the policy architecture seems straightforward and enables PG to implicitly target an MI metric associated with good coordination.
- The authors provide both a theoretical analysis of InfoPG (although, again, see Weaknesses below) and empirical demonstration of its effectiveness.
- InfoPG accounts for the possibility that not all agents provide meaningful communication.

# Weaknesses
- The abstract and introduction talk about bounded rationality, but that concept doesn't seem relevant thereafter.
- Much of the paper is devoted to demonstrating the *implicit* connection between MI of agents' actions and the InfoPG policy formulation. This comes across as something of a *post hoc* motivation for the setup. It makes good enough sense why MI and coordination are related, but (as the paper points out) treating MI maximization as the objective is potentially short sighted. In sum, I question if MI is the best lens to motivate/understand this approach.
- Good performance is encouraging, but the most relevant analyses for validating the intuitions of the method are largely ignored in the main text.
- From the description in A.9, it sounds like the connection between *k*-level reasoning and the actual architecture implementation is misleading. It seems more appropriate to think of the agents' policies as one large joint policy. Please justify this connection and/or clarify the mechanisms by which agents' actions become correlated. From my understanding, InfoPG (with this architecture) would not cause the MI between the 3rd and 2nd piston to actually change over time in the BGP experiments -- that is, learning would not bring about any reduction in their MI as the main text describes.

# Questions/Comments
- It is hard to understand the aspects of InfoPG that set it apart from prior work. It may help to reconsider where MOA and PR2 are first detailed.
- In the Section 3 paragraph about MI, you write: ```In our work, X and Y are policy distributions of two interacting agents.``` This is vague. Please be more clear about what the "policy distributions" are. I'm assuming it's the action distribution of each policy, but the paragraph is not very concrete.
- Consider making it a bit more clear what "InfoPG" refers to. The k-level encoding/decoding policy definition? Something to do with the objective? Perhaps Section 4.3 is sufficiently clear, but just as a note I got a little bit confused about what "InfoPG" is meant to describe.
- For the bottom row of Figure 2, how are you estimating MI? It would be good to state that in the paper. Also, what makes this an "ablation" study?

**Summary Of The Paper:**

This work introduces InfoPG for learning coordination in fully decentralized multi-agent games. InfoPG models agent policies following *k*-level reasoning. The authors present a theoretical analysis to show that, under InfoPG, policy gradient optimizes a form of mutual information related to coordination. Empirical demonstrations show that InfoPG enables better performance than conceptually related baselines, with some demonstration of how InfoPG better adapts to a version of a Byzantine Generals Problem.

**Summary Of The Review:**

My starting recommendation is: weak reject.

I hope the authors can use the rebuttal period to correct any misunderstandings that may contribute to my lower score.

However, assuming I have understood things accurately, I am primarily concerned with the notion that InfoPG is a genuinely decentralized approach. It seems to me that the policy is centralized because of how the communication policy works. I wonder how much the benefit of InfoPG simply comes from the fact that each agent naturally gets more information about the state of the world. More generally, I worry that the work misrepresents itself, using phrases like "communication" and "*k*-level reasoning" where they may not be appropriate. I am interested in how the authors defend these choices and characterizations.

I also cannot escape the sense that the MI lens is somewhat *post hoc.* It may be possible to show that other MARL algorithms implicitly target MI, if it is indeed a corollary of coordination. Bottom line: a lot of the paper is spent on the theoretical connections between InfoPG and an abstract metric; I think that space would have been better spent on empirical analysis. Again, if I have overlooked the importance of (this form of) MI in these task settings, I hope the authors will correct me.

---

> ### Author Response · Authors · 2021-11-14
> **Official Response to Reviewer SyxF -- PART 3**
>
> ### Summary of the Review
>
> We appreciate the reviewer’s constructive discussions that helped us improve our manuscript. We provided detailed discussions for all the points and made appropriate edits respective to the comments.
>
> Regarding the MI being somewhat post hoc and other MARL algorithms’ ability to implicitly increase MI we would like to add that this heavily depends on what mutual information you intend on maximizing. Since MI can only be drawn between two probability distributions, it matters that what probability distributions are being used to draw MI. With our $k$-level communication we directly seek to maximize the MI between agents’ action distributions to affect their coordination. Take TarMAC (or other similar approaches) for instance, which also shares information among agents which isn’t necessarily an action distribution, and since the entire team reward is being maximized without considering each individual agent, it is not concretely defined that each agent will be maximizing mutual information of their action distributions with one another. What we do instead is that we provide strict guarantees of per-agent MI maximization by explicitly considering action-conditional policies for each agent.
>
> We would be happy to address any further questions, comments, or modifications that the reviewer may believe are needed. Thank you!

---

> > ### Comment · Reviewer_SyxF · 2021-11-18
> > **Update on review**
> >
> > Thank you for taking the time to provide such detailed responses to my concerns and questions. I am now clearer on the contributions of the work. Overall, my takeaway is that I did have some misunderstandings which contributed to a lower evaluation, and I will raise my score to reflect that.
> >
> > I am still somewhat unsure about the value added by analyzing InfoPG through the lens of MI. (I will use "MI" to refer to the form of MI relevant to this context.) Perhaps *post hoc* was the wrong criticism. My thinking is this...
> > There is evidence that MI measures or correlates with coordination, as you discuss. Others have used that to motivate explicit MI based objectives. You make the case that MI maximization is not always beneficial. Your analyses indicate that InfoPG will *implicitly* increase MI when it is beneficial and decrease it otherwise. The value of this result depends on whether we regard MI as some gold standard of coordination. It is my perspective that MI is instead a more abstract metric.
> > Nevertheless, I suppose there is enough prior work to justify the attention MI gets in your work. And your argument in Part 1 does help to answer the question of why we should care about MI in this context.
> >
> > Since InfoPG optimizes MI *implicitly* by virtue of its architecture, the architecture seems to be the backbone of the innovation. In that case, it is perhaps best validated by its superior performance. I am less convinced that concepts of bounded rationality and iterated reasoning are relevant. Rather, I think it is more parsimonious to simply view the choice of $k$ as a hyperparameter of network depth. Evaluating whether that depth imparts additional stages of reasoning (i.e. to resemble bounded rationality) requires more targeted analysis. If memory serves, Wen et al. (2019) look for signatures of this. If you added more concrete analyses along those lines, or ditched the bounded rationality portion of the narrative, I would view the paper with less skepticism.
> >
> > Thanks again for your thoughtful responses. As before, if it seems I have misunderstood a point, I welcome the correction.

---

> > > ### Author Response · Authors · 2021-11-21
> > > **Update on Response to Review**
> > >
> > > We sincerely appreciate the reviewer for taking the time to respond and their constructive discussions that helped us improve our manuscript further.
> > >
> > > We want to start our response by stating that we agree with the reviewer's point that further targeted analysis can better justify and help to show if our architecture actually resembles bounded rationality assumption and our iterated reasoning is in fact similar to k-level thinking. As such, we performed a detailed qualitative analysis of our k-level policies to investigate the effects of varying levels of bounded rationality in two of our most challenging domains, the SC2 and the Multiwalker. We added this new analysis to the manuscript, in the Supplementary Results section (section A.7), as the subsection A.7.2 (pages 21-23). While we briefly summarize our analysis for SC2 here, we would like to invite the reviewer to please check this new section and our qualitative analysis of the bounded rationality.
> > >
> > > In our analysis, we compare the learned policies by InfoPG at convergence in the SC2 domain with $k=0$ and $k=1$. In SC2, agents are positively rewarded for shooting and killing enemy agents, and are negatively penalized for getting shot at. Therefore, a locally optimal strategy is to run away from the enemy team to avoid any negative penalties, while a globally optimal strategy is to kill and eliminate all the enemy agents to achieve high positive rewards. At $k=0$ of the rationalization hierarchy, the fully naïve and non-strategic level-$0$ agents choose actions while completely disregarding other agents actions (i.e., have zero-order beliefs). As such, for a level-$0$ policy, we expect to observe that agents simply run away from the enemy to avoid getting shot at, since a single agent does not believe (zero-order belief) it can overcome the enemy team. We demonstrate, as shown in the newly added Figure 5 (page 22), that the naïve level-$0$ agents expectedly only learn to run away to avoid the negative penalties of getting shot at. This fleeing behavior allows agents to maintain a reward of zero, as shown in Figure 2, indicating successful escape and convergence to the locally optimum solution.
> > >
> > > At level $k=1$, each agent is now more sophisticated and believes that the other agents have a level-$0$ policy and takes actions according to that. In this case, we observe a vastly different behavior. Again, we demonstrate in the newly added Figure 5 that agents now learn more strategic policies to work together to eliminate the enemy team and achieve high positive rewards. Through reasoning their level-$1$ actions based on their teammates level-$0$ action, InfoPG agents learn a sacrificial technique of exposing one agent as bait, which allows the team to converge to the globally optimum solution of killing the entire enemy team. This is also reflected in Figure 2, where the $k=1$ InfoPG achieves the highest cumulative rewards.
> > >
> > > We also compare our level-$k$ policies for $k=1$, $k=2$, and $k=3$ in the Multiwalker domain and provide a similar subtle analysis of the effects of varying levels of bounded rationality (presented in Section A.7.2 on page 22 and the newly added figure 6 on page 23).
> > >
> > > As a results of our analysis, we believe that the our $k$-level architecture qualitatively resembles bounded rationality and our iterated reasoning for decision making in fact mimics a $k$-level thinking strategy. We do agree that further quantitative and more targeted analysis can improve this conclusion. The matrix games analysis in Wen at al. (2019) also presents a similar qualitative analysis (the "smart path" taken by PR2 versus the path taken by others) where the authors show their approach can converge to global optimum rather than a local one. Wen at al. (2019) additionally quantitatively shows the results of this qualitative analysis in optimization graphs. We tried to qualitatively show and discuss a similar behavior in InfoPG. We leave conducting such quantitative analysis for future work however, due to limited rebuttal time.
> > >
> > > Once again, we thank the reviewer for their time and dedication. We would be happy to further address any questions or concerns.
> > >
> > > Thanks!

---

> > > > ### Comment · Reviewer_SyxF · 2021-11-22
> > > > **Reply**
> > > >
> > > > Thank you for taking the time to include these analyses. I think they do add to the characterization of your method and, at least in part, connect it to its conceptual underpinnings.
> > > >
> > > > I do realize that there are limits to how the paper can change during reviews, but I have to judge the paper as a whole. So, while I think these qualitative analyses add value, they are still buried quite deep in the paper, and I think it may still be premature to interpret $k$ as the levels of thinking based on the presented evidence.
> > > >
> > > > In sum, I still lean more towards a 6 than towards an 8. I'll await potential input from the other reviewers before finalizing that decision.

---

> > > > > ### Author Response · Authors · 2021-11-22
> > > > > **Authors' Response**
> > > > >
> > > > > Thank you very much. We again appreciate your time and constructive feedback. We look forward to following up after other reviewers have potentially responded. Thank you!

---

> ### Author Response · Authors · 2021-11-14
> **Official Response to Reviewer SyxF -- PART 2**
>
> - Thank you. InfoPG agents’ policy should not be considered as one large policy and here we clarify our mechanism:
>
> &emsp;&emsp;We equip each agent with a local state-conditional policy, which takes a state and produces an action-distribution, and a local action-conditional policy, which recursively integrates the communicated action-distributions from adjacent agents into the current action-distribution for k-steps. Note that, although there’s communication in our approach, agents share with their neighbors only high-dimensional action embeddings rather than their actual selected action, and therefore, the receiving agent still needs to “reason” about the sender’s action using the acquired information. This setup however is not a large joint policy, because (1) each agent’s action-conditional policy integrates other agent’s action-distributions if those agents are within a delta-sized neighborhood of the time-varying communication graph (for instance in StarCraft or Pistonball) and (2) each agent has its own local optimizer that optimizes its individual policies using its individual rewards, rather than a single optimizer that optimizes the sum of the objectives of all agents.
>
> &emsp;&emsp;Having a single optimizer is characteristic of true centralized or CTDE joint policies like CommNet, TarMAC, DIAL, etc. Consider Multiwalker; this experiment has two agents that are on their own, each trying to keep their own balance while supporting a package. If the system were a centralized joint policy, then the objective would be to maintain overall stability, and a single optimizer would be employed. In our case, each walker generates its own behavior to optimize for local stability. In fact, we have sometimes observed that the left-most walker learns to generate forward momentum, and walk more quickly than the front walker, which learns to walk more slowly, but maintain the stability of the package. Or consider StarCraft, each agent is dynamically moving and can only communicate with other teammates when they are within range. We would like to note that this definition and setup fall exactly in the realm of decentralized learning and Dec-POMDPs pursuant to (Zhang et al., 2018; Oliehoek & Amato, 2016; Melo et al., 2011).
>
> &emsp;&emsp;With respect to the BGP scenario, agents are introduced with no pre-conceived notion that agent 2 is faulty, so MI might actually be positive between agent 3 and agent 2 at the beginning of the experiment. Conceptually, agent 3 does not know that it should not listen to the gibberish actions of agent 2 at the beginning of the experiment. However, our experiments show that after training with Adv. InfoPG, which allows for a decrease in MI based on the cooperativity of agents (discussed in section 4.6), the MI between agent 2 and agent 3 converges to zero (evidenced by the non-changing distribution of agent 3 after communicating with agent 2). Conversely, agent 3 learns to communicate with agent 4, evidenced by the changed action-distribution after communicating with agent 4, because agent 4 is not producing gibberish actions and coordinating with this agent benefits agent 3.
>
> ### Questions/Comments:
> - Thanks for pointing out. We added further discriminating details of PR2 and MOA to page 7, Section 5, the Baselines subsection. We also revised the text in Section 2 slightly to differentiate the differences more clearly between InfoPG and the prior works MOA and PR2. Hope this is a satisfying revision.
>
> - Good point! Thanks. $X$ and $Y$ refer to distributions over actions given a specific state. We revised the text on the bottom of page three to include this more concrete definition.
>
> - Thank you. InfoPG is mainly meant to refer to using a $k$-level action conditional policy during a Policy Gradient in a decentralized reward setting that implicitly increases mutual information among agents with respect to the observed cooperativity and leads to improved coordination. We revised the text to further describe this point in the Introduction Section (page two, second paragraph, the eighth line under the Contributions).
>
> - We are estimating MI as the average between the lower and upper bounds defined in equations 6 and 9, respectively. We added this information to the text. We are also calling this subsection the “Mutual Information Variation Analysis” rather than “Mutual Information Variation Ablation Study”. Thanks!

---

> ### Author Response · Authors · 2021-11-14
> **Official Response to Reviewer SyxF -- PART 1**
>
> We appreciate the reviewers time and constructive feedback and discussions. Below we respond to all the points made one-by-one.
>
> ### Strength
> Thank you for pointing out the strength in our work, including the clarity in presentation.
>
> ### Weaknesses
> - In the introduction and abstract, we mention the idea of bounded rationality to motivate our recursive policies; This idea is particularly relevant to the empirical evaluation of our algorithm. As mentioned in section 4.2, the equilibrium condition for bounded rationality is when the performance of the policy at $k+2$ matches the policy at $k$. In Pistonball and Multiwalker, we see performance improvements when increasing $k=1$ to $k=2$, while we observe relatively similar performances for $k=2$ and $k=3$. This indicates that InfoPG can benefit from assuming bounded rational agents and deeper iterated reasoning for decision making. We leave the rationalization level $k$ to be a hyper-parameter for the experimenter to set due to its application dependency. This discussion is provided in Section 6: Deep Reasoning for Decision Making: Evaluating $k$-Level InfoPG.
>
> - Thank you for noting this important point. Our objective seeks to maximize individual rewards (as opposed to a shared reward), which we mathematically show will implicitly maximize mutual information at the same time. Now, we ask “Is mutual information maximization just a side benefit to maximizing individual rewards when agents have a k-level policy?” The answer to this question relies upon examining when agents should and should not collaborate (or share meaningful information) in a decentralized reward setting. If mutual information is defined as information gain for conditional policies, we posit: for two agents, A and B, if B provides information to A and that leads to A performing an action that leads to higher rewards than if A performed its action without B’s influence, then A should positively reinforce “listening” to information acquired from B. Otherwise, what individual benefit does A have to listen to B, if A never concretely gains rewards by conditionalizing on B?
>
> &emsp;&emsp;Contrary to previous works in MI maximization (Kim et al., 2020; Wang et al., 2019; Jaques et al., 2019) which externally regularize MI, regardless of individual reward performance, we theorize that, in a decentralized setting, there is an inherent tie between individual performance and conditional information gain. While agents can maximize individual rewards without conditional information gain, (which is just NC-A2C), we show that InfoPG’s ability to reinforce conditional information gain during non-negative reward timesteps leads to cooperative performance and therefore better overall performance to baselines in our cooperative settings (Pistonball, Pong, Multiwalker, StarCraft).
>
> &emsp;&emsp;There are two perspectives we consider 1) InfoPG - non-negative mutual information maximization, which means agents will only “listen” to the information of others when they achieve positive rewards from that information 2) Adv. InfoPG - mutual information fluctuates both positively and negatively with instantaneous performance. We introduce Adv. InfoPG because we seek to tackle the BGP scenario, which is a specific collaborative scenario with a fraudulent agent that other agents should learn not to listen to.
>
> &emsp;&emsp;Therefore, in summary, we believe that our relation to MI should not be considered post-hoc, nor a direct optimization objective per previous literature. Prior work states that encouraging conditional information gain (MI) leads to cooperative performance, and we state that it is more intuitive to maximize conditional information gain (MI) when positive individual rewards are experienced. We would be happy to include this discussion as a Takeaway in our supplementary results, if the reviewer thinks it would be helpful and appropriate.
>
> - We would like to point out that besides showing the superior performance of InfoPG compared to baselines, in section 6, we try to address and validate main intuitions and claims made in out method. Particularly, we investigate the MI variations and achieving higher MI compared to the MOA baseline in the Mutual Information Variation Ablation Study sub-section of section 6. We investigate the bounded rationality intuition and effects of deeper levels of $k$ for decision making in the Deep Reasoning for Decision Making subsection of section 6. And in the last subsection of Section 6, The Fraudulent Agent Experiment: Regularising MI, we seek to analyze the intuitions behind proposing Adv. InfopG and modulating MI, as discussed in section 4.6. Due to page restrictions, we provide further in-depth policy interpretation and analysis for the BGP scenario within the appendix since we believe that those could be considered as supporting results. Nevertheless, we would be happy to include any further analysis that the reviewer may think should be covered in the main text.

---

### Official Review · Reviewer_4Fxp · 2021-11-04

**Correctness:** 3
**Technical Novelty And Significance:** 3
**Empirical Novelty And Significance:** 2
**Recommendation:** 3
**Confidence:** 4

**Details Of Ethics Concerns:**

While any approach that actively deals with information sharing and agent modeling introduces some risk of being abused by malicious actors, I am not sure this study poses substantially more risk than most work in the multiagnet RL field. Having said that, it is always good to keep those considerations in mind and have an expert examine them more carefully.

**Main Review:**

While there are plenty of interesting ideas in this paper, and this kind of approach to multiagent learning that relies both on information theory and game theory seems to hold great promise, I find there are multiple issues with the paper that must be addressed for it to be of greater utility to the ML and RL communities.

- First and foremost, it seems the authors omit from their literature review (and subsequently from their consideration of eligible baselines) works relating to two heavily relevant bodies of work: work on hierarchical learning in Dec-POMDPs (for instance Amato et al. 2019), and work on learning to communicate in multiagent RL (for instance Foerster et al. 2016, or Sheikh and Boloni 2019).

- At the core, what the paper seems to propose, ultimately, is a framework for jointly learning to communicate *and* act simultaneously, which is both interesting and useful, but again, certain related work seems to be missing in that regard (consider Ghavamzadeh and Mahadevan 2004 as a relatively early - but still related - example).

- A potentially critical consideration which the paper does not seem to explicitly address in its theoretical analysis is that it's unclear from the setting and the (rather muddled, more on that later) formulation of the algorithmic approach how the proposed framework avoids the plague of decentralized learning in multiagent RL, which is nonstationarity. In fact, it almost seems baked into the solution that if each policy learned is the best response to the policies at the lower level, then nothing is necessarily guaranteed to converge. There's also the risk that implicitly the extent of communication baked into the framework renders the approach analogous to single agent learning (theory of mind taken to its extreme), and it is unclear to me from the paper to what extent this might be the case (there is a proof in the supplementary material that eq. 2 does converge, but because the entire framework is so muddled - more on that later - it's not clear how that resolves the stability issue for the larger framework, and also, one cannot rely on supplementary material to establish key aspects of their approach).

- in practice the communication setup proposed in the paper, as much as I can understand it, seems rather contrived and unrealistic. This may not be a disqualifying aspect of the approach - we often need to make certain simplifying assumptions to establish initial results and study novel approaches, but it does raise the question of how well this approach will hold up if certain constraints on communication are imposed.

- Unfortunately my biggest issue with the paper is lack of clarity. Terms are introduced without sufficiently elucidating their meaning in the specific context of the paper (even as simple an example as "rationalizability" right in the abstract), Section 4.2 which should be the backbone of the paper is very confusing to read, Section 4.3 was even harder for me to follow, and ultimately by the end of Section 4, before discussing the empirical evaluation, while I retained a high level mental picture of the general outline of the proposed approach, I could not technically describe how the actual learning framework was designed, how the hierarchical approach works in practice, and how the framework is trained (the supplementary material helps somewhat, but not enough, and again, authors should not rely on supplementary material to present any key aspect of their work, they are called "supplementary" for a reason).


**Summary Of The Paper:**

This paper discusses a hierarchical, iterated reasoning scheme for learning policies conditioned on the other agents' policies at that stage,  maximizing both the mutual information between agent policies and agent rewards. The authors leverage cognitive hierarchical theory wherein each agents has multiple levels of abstraction and reasoning about the actions other agents might take, and take a game-theoretic approach to policy learning in which each subsequent policy is the best response based on other agents' previous policies. The authors proceed to show this approach yields results which outperform several baselines in complex environments such as StarCraft II and multiwalker.

**Summary Of The Review:**

This paper has interesting ideas but is lacking in how it anchors its contribution to related work and compares against it. More glaringly, this papers suffers from substantial lack of clarity both in terms of its theoretical grounding, and more critically, in terms of how it presents its core framework, how it is designed, and how it is applied. I believe it should be revised substantially before it is suitable for publication at a venue such as ICLR.

---

> ### Author Response · Authors · 2021-11-14
> **Official Response to Reviewer 4Fxp -- PART 2**
>
> - We strongly disagree based upon the communication formalisms of Dec-POMDPs pursuant to Zhang et al. (2018), Oliehoek & Amato (2016) and Melo et al. (2011). Our communication messages include an action distribution (i.e., represented as a limited-length vector) which are only communicated with another agent when within a delta-distance. Such local communication is a property in the formalism of the aforementioned prior work.
>
> - We appreciate the reviewer’s concern regarding the clarity of our manuscript. We use the term “rationalizability” by its straightforward English meaning to describe an agent’s ability to reason about its action decisions. We also tried not to leave any terms unexplained. We believe our iterative communication and learning framework are well described in Section 4 as well as through a descent amount of supplementary information that we provided, such as the InfoPG pseudocode with line-by-line description as well as the training and execution details for each specific domain. We provide such information for interested individuals in the supplementary as we believe that they play a supporting role for the key contributions in our work. We hope that the material provided in our appendix are a representation of our thoroughness and comprehensiveness in our investigations. We would be happy to address reviewer’s concern regarding any other unexplained term they may be referring to. Can the reviewer please be more specific about what aspects of Sections 4.2 and 4.3 were confusing? We would be happy to address those here and in the paper.
>
> ### Summary of Responses
> - [Points 1—2] We appreciate the reviewer for pointing out the related work on hierarchical MARL for coordination. We revised our related work section to cover the prior work on hierarchical MARL for coordination, including the works mentioned by the reviewer in the first two points. We compare our fully decentralized work against recent, conceptually related state-of-the-art that share our decentralized learning architecture.
>
> - [Points 3—4] We responded to the reviewer’s two follow-up questions regarding convergence of policies in our $k$-level paradigm and provided references for our claims.
>
> - [Point 5] We provide a significant amount of well-justified theoretical grounding for the key aspects of our work; our core framework is concretely and concisely described within Section 4 and the implementation and training subtleties are provided thoroughly in the appendix for interested individuals. Nevertheless, we would like to ask the reviewer to kindly point out specifically where our manuscript lacks theoretical grounding and clarity in presenting its core ideas?
>
> ### Flag For Ethics Review
> We would be grateful if the reviewer could either retract their flag or point to specific papers that argue that using implicit mutual information for multi-agent coordination poses a sincere, ethical risk.

---

> ### Author Response · Authors · 2021-11-14
> **Official Response to Reviewer 4Fxp -- PART 1**
>
> We appreciate the reviewers time. Below we respond to all the questions one-by-one.
>
> - First, we did discuss Foerester et al. (2016) and other related works in Section 2, first paragraph, second line: “… to promote coordination (Foerster et al. 2016; Das et al. 2018; …” Finally, we did not consider these works as eligible baseline because they utilize a CTDE training framework while InfoPG is implemented on a fully decentralized architecture. We instead benchmark against more conceptually related works for a fair comparison. &nbsp;&nbsp;&nbsp;&nbsp;&nbsp;&nbsp;&nbsp;&nbsp;&nbsp;&nbsp;&nbsp;&nbsp;&nbsp;&nbsp;&nbsp;&nbsp;&nbsp;&nbsp;&nbsp;&nbsp;&nbsp;&nbsp;&nbsp;&nbsp;&nbsp;&nbsp;&nbsp;&nbsp;&nbsp;
>
>      Second, we omitted these works as they do not serve as relevant baselines. The work by Amato et al. (2019) assumes access to options (i.e., macro actions) whereas we are learning a policy over low-level actions. The work by Foerster et al. (2016) and others does not serve as a relevant baseline because those works perform centralized training, whereas ours is fully decentralized both in training and execution. The work by Sheikh and Boloni 2019 is constructed on MADDPG, which again benefits from a CTDE framework and thus is not relevant to our architecture.
> &nbsp;&nbsp;&nbsp;&nbsp;&nbsp;&nbsp;&nbsp;&nbsp;&nbsp;&nbsp;&nbsp;&nbsp;&nbsp;&nbsp;&nbsp;&nbsp;&nbsp;&nbsp;&nbsp;&nbsp;&nbsp;&nbsp;&nbsp;&nbsp;&nbsp;&nbsp;&nbsp;&nbsp;&nbsp;
>
>      We will include in our related works references to these additional citations though for completeness. Thank you for the suggestion.
>
> - Thanks for pointing out this work. We revised our related work section to cover the prior work on hierarchical MARL for coordination, including the works mentioned by the reviewer in this and the previous point. We hope this revision is satisfactory.
>
>
> - $k$-Level Reasoning is a recursive approach but is not necessarily infinitely recursive. The postulate of the bounded rationality is that there exists an optimal $k$ where further recursive reasoning does not further improve the policy at $k$. The same logic applies with the way our $k$-level policies are constructed. To show this assertion is true, we examined $k=1, 2$ and $3$ across our evaluations domains and provided empirical evidence of convergence behavior with respect to increasing $k$. In general, we state that $k$-level reasoning can generate better collaborative behavior, which we justify through MI maximization. Further, by following the steps of two strong prior work (Bhatnagar et al., 2009 and Zhang et al., 2018) we theoretically prove the convergence of each agents’ individual policies for a fixed $k$ through a two-timescale approach. As such, if an agent’s individual policy is converging, so is the Best-Response policy to this policy. The optimal $k$ is a hyper-parameter left up to the researcher to discover; however, there is strong literature stating that optimal $k$ should exist (Gershman et al. (2015); Ho & Su (2013)).
> &nbsp;&nbsp;&nbsp;&nbsp;&nbsp;&nbsp;&nbsp;&nbsp;&nbsp;&nbsp;&nbsp;&nbsp;&nbsp;&nbsp;&nbsp;&nbsp;&nbsp;&nbsp;&nbsp;&nbsp;&nbsp;&nbsp;&nbsp;&nbsp;&nbsp;&nbsp;&nbsp;&nbsp;&nbsp;
>
>      Furthermore, we disagree that this framework is analogous to single agent learning. We allow each agent to have only a local state, observation, and action-conditional policy for which communication only occurs when other agents come within communication range. Therefore, the policies might be instantaneously joint when agents are within range, but, when considering the entire trajectory of the team, the agents’ policies are not continuously joint. Thus, we believe that it is more appropriate to think of agents’ policies as individual policies that optimize individual objectives but tend to communicate when other agents come in range, and this definition falls exactly in the realm of decentralized learning pursuant to (Zhang et al., 2018; Oliehoek & Amato, 2016; Melo et al., 2011). We employ individual optimizers to optimize individual rewards per agent, whereas traditional centralized architectures like CommNet, TarMAC, and DIAL use a single optimizer to optimize the sum of the individual objectives of each agent.

---

> ### Author Response · Authors · 2021-11-19
> **Follow-up on the Rebuttal Discussion**
>
> Please let us know if we can provide any more information in support of the paper's acceptance. We revised the paper to address the missing references and answered the questions you mentioned. If our rebuttal has addressed your concerns, we kindly ask the reviewer might consider please increasing the reviewer's score? Otherwise, we would be happy to continue discussing any remaining items! Thank you again for your review.

---

### Public Comment · ~Xihuai_Wang1 · 2022-04-01
**Questions about the bound**

I have some questions about the theories in this paper.
1. How to define a MAP action formally? Can I understand it as $argmax_{a^{i}} \pi^{i}(\cdot|a^{j})p(a^{j})$?
2. How is the last but one inequality in equation (8) derived? Why can the MAP action be selected as the particular one that satisfies the inequality? If my understanding in *1.* is right, the inequality should be less than or equal to ($\leq$) instead of greater than or equal to ($\geq$)? Can you give some proofs on the existence of such particular MAP action?

Hoping for your replication.

---

> ### Public Comment · ~Esmaeil_Seraj1 · 2022-04-11
> **Response to Question Regarding the Bound**
>
> Thanks. Please find the responses below:
>
> 1- We define the MAP action as the most likely action (i.e., highest probability action) w.r.t InfoPG objective. This is:
>
> $$
> argmax_{a^{i, (k)}}(\Pi_i^k\pi_{com}^i(a^{i, (k)}\vert a^{i, (k-1)},a^{j, (k)}).\pi_{enc}(a^{i, (0)}\vert o^i).\pi_{enc}(a^{j, (0)}\vert o^j))
> $$
>
> 2- That's incorrect. The inequality should be greater-than-or-equal-to ($\geq$) since the left side, i.e., expansion of MI, is strictly non-negative, while the right-side is a multiplication between a negative, $log(\pi)$, and a positive term, $\pi$. The proof is relatively straight-forward. For a sketch proof, see below:
>
> $$
> \rightarrow \frac{1}{|A|}[\sum_{a_j}\sum_{a_i}\pi^i(a^i|a^j)log(|A|\pi^i(a^i|a^j))]
> $$
> $$
> = \frac{1}{|A|}[\sum_{a_j}\sum_{a_i}\pi^i(a^i|a^j)log(\pi^i(a^i|a^j)) + \pi^i(a^i|a^j)log(|A|)]
> $$
> $$
> = \frac{1}{|A|}[\sum_{a_j}\sum_{a_i}\pi^i(a^i|a^j)log(\pi^i(a^i|a^j))] + \frac{log(|A|)}{|A|}[\sum_{a_j}\sum_{a_i}\pi^i(a^i|a^j)]
> $$
> $$
> = \frac{1}{|A|}[\sum_{a_j}\sum_{a_i}\pi^i(a^i|a^j)log(\pi^i(a^i|a^j))] + \frac{log(|A|)}{|A|}
> $$
> $$
> \geq \frac{1}{|A|}[\sum_{a_j}\sum_{a_i}\pi^i(a^i|a^j)log(\pi^i(a^i|a^j))]
> $$
> $$
> \geq \pi^i(a^i|a^j)log(\pi^i(a^i|a^j))
> $$
>
> where eventually the MAP action is selected particularly that satisfies this inequality.

---

### Decision · Program_Chairs · 2022-01-20

**Decision:**

Accept (Poster)

**Comment:**

The paper proposes a method for decentralized learning of cooperative games by maximizing the mutual information between the agents. The paper is novel and interesting and well evaluated.

Prior to the rebuttal, most of the reviewers saw presentation as the biggest weakness. Specifically, it was not clear what InfoPG refers to, and how it is related to the mutual information. During the rebuttal the authors cleaned up the misunderstandings around the presentation and provided a detailed analysis in the Appendix.

While the author responses provided helpful clarification and analysis, the authors should revise the paper holistically to remove unnecessary terminology and connections, and bring the analysis in the main text.